# Phylogenomic curation of Ovate Family Proteins (OFPs) in the U's Triangle of *Brassica* L. indicates stress-induced growth modulation

**Muhammad Shahzaib**[1,2☯], **Uzair Muhammad Khan**[2☯], **Muhammad Tehseen Azhar**[3], **Rana Muhammad Atif**[2,3], **Sultan Habibullah Khan**[1,2], **Qamar U. Zaman**[4,5], **Iqrar Ahmad Rana**[1,2]*

**1** Centre of Agricultural Biochemistry and Biotechnology, University of Agriculture, Faisalabad, Faisalabad, Punjab, Pakistan, **2** Centre for Advanced Studies in Agriculture and Food Security, University of Agriculture, Faisalabad, Faisalabad, Punjab, Pakistan, **3** Department of Plant Breeding and Genetics, University of Agriculture, Faisalabad, Faisalabad, Punjab, Pakistan, **4** Hainan Yazhou Bay Seed Laboratory, Sanya Nanfan Research Institute of Hainan University, Sanya, China, **5** College of Tropical Crops, Hainan University, Haikou, China

☯ These authors contributed equally to this work.

* iqrar_rana@uaf.edu.pk

## Abstract

The Ovate Family Proteins (OFPs) gene family houses a class of proteins that are involved in regulating plant growth and development. To date, there is no report of the simultaneous functional characterization of this gene family in all members of U's Triangle of *Brassica*. Here, we retrieved a combined total of 256 OFP protein sequences and analyzed their chromosomal localization, gene structure, conserved protein motif domains, and the pattern of *cis*-acting regulatory elements. The abundance of light-responsive elements like *G-box*, *MRE*, and *GT1 motif* suggests that OFPs are sensitive to the stimuli of light. The protein-protein interaction network analysis revealed that *OFP05* and its orthologous genes were involved in regulating the process of transcriptional repression through their interaction with homeodomain transcription factors like *KNAT* and *BLH*. The presence of domains like *DNA binding 2* and its superfamily speculated the involvement of OFPs in regulating gene expression. The biotic and abiotic stress, and the tissue-specific expression analysis of the RNA-seq datasets revealed that some of the genes such as *BjuOFP30*, and *BnaOFP27*, *BolOFP11*, and *BolOFP10* were highly upregulated in seed coat at the mature stage and roots under various chemical stress conditions respectively which suggests their crucial role in plant growth and development processes. Experimental validation of prominent *BnaOFPs* such as *BnaOFP27* confirmed their involvement in regulating gene expression under salinity, heavy metal, drought, heat, and cold stress. The GO and KEGG pathway enrichment analysis also sheds light on the involvement of OFPs in regulating plant growth and development. These findings have the potential to serve as a forerunner for future studies in terms of functionally diverse analysis of the OFP gene family in *Brassica* and other plant species.

**Data Availability Statement:** All relevant data are within the paper and its Supporting Information files.

**Funding:** This work was supported by the Higher Education Commission of Pakistan (HEC) through its grant of Precision Agriculture and Analytics Lab (PAAL) under the National Centre in Big Data and Cloud Computing (NCBC). The funders had no role in study design, data collection, and analysis, the decision to publish, or the preparation of the manuscript.

**Competing interests:** The authors have declared that no competing interests exist.

## 1. Introduction

The *Brassica* genus holds crops of great agricultural importance. Among its members, *B. rapa*, *B. nigra*, *B. oleracea*, *B. juncea*, *B. napus*, and *B. carinata* are the six important species due to their interesting course of complex interdependent evolution. The evolutionary relationships of these members were first described by the U's Triangle model which places *B. oleracea*, *B. nigra*, and *B. rapa* as ancestors that procreated allopolyploid *B. juncea*, *B. napus*, and *B. carinata* [1, 2]. The whole genome sequencing (WGS) of all these members has been completed by utilizing a *de novo* assembly of Hi-C, PacBio, Nanopore, and Illumina sequencing modules [3]. The availability of genomic data for all the members of U's Triangle made simultaneous comparative and functional genomic analysis possible.

Ovate Family Proteins (OFPs) were first discovered in *Solanum lycopersicum* L. (Tomato) over a century ago and were first cloned in 2002 [4, 5]. In the original studies, OFPs were found to have a crucial role in determining the pear-type elongated shape of tomato and peach fruit [6, 7]. They reside in a conserved *OVATE* domain of 70 amino acid residues in length. Shared *OVATE* domains have been characterized in plant species such as *Arabidopsis thaliana*, tomato, and rice [8]. Along with the aforementioned role, OFPs were also characterized to have roles in optimizing plant growth and development processes through the process of transcriptional repression [4]. For example, in the case of the tubers of *Raphanus sativus* (Radish), the *RsOFP2.3* gene mediates their extent of elongation [9]. Another study on *B. napus* found that the *BnOFP13_2* was involved in controlling the frequency of seeds per silique [10]. In *A. thaliana*, *AtOFP05* was involved in suppressing the activity of the BELL-KNOX TALE homeodomain transcription factor which controls the cell-fate switch during embryonic development [11]. Similarly, *AtOFP01* was involved in performing numerous functions like working alongside the *ATH1* factor to regulate stem growth and flowering when alone affecting DNA repair, male gametogenesis, and activity of pollens. In conjunction with *AtOFP04*, *AtOFP01* also influences the structural development of secondary cell walls [8, 12–15].

*Oryza sativa* also shows the aforementioned BELL-KNOX TALE suppression mediated by *OsOFP02* along with the control of leaf and seed shape, plant height, the vascularity of stems, and biosynthesis of lignin [16]. The regulation of fruit shape in *Prunus persica* (Peach) is mediated by *PpOFP1* [7]. Moreover, *MaOFP01* is involved in fruit ripening in *Musa paradisiaca* (Banana), *CmOFP13* in the fruit shape *Cucumis melo* (Melon), *CsOFP12-16c* in the silique development of *Cucumis sativus* (Cucumber), *CaOFP01* and *CaOFP20* in the fruit shape and length of *Capsicum annuum* (Sweet And chili pepper), and lastly *GhOFP04* in the fiber development of *Gossypium hirsutum* (Cotton) [17–23]. All these studies suggest that OFPs have a main supporting role in the modulation of different aspects of plant growth and development.

The diversity in fruit morphology, transcriptional repression, responses to biotic and abiotic stresses, and other important aspects of plant growth and development processes are also controlled by several OFP-associated molecular regulators [24–45]. In the *Brassica* species of the U's triangle, the role of OFPs has been poorly understood and to our knowledge, only a few studies have been reported [10, 46]. Characterizing their specific roles in influencing the underlying mechanisms that mediate the plant growth and development processes may provide the future basis for agronomic crop development and improvement programs. In the present study, we have studied the spatiotemporal distribution of OFPs across the U's Triangle members. This comparatively functional genomic analysis would provide us with a better understanding of the role of OFPs.

## 2. Materials & methods

### 2.1 Identification of OFP genes

In this study, the identification of the OFP genes in *Brassica* was performed using the query input OFP gene sequences derived from the *Arabidopsis thaliana*'s native genomic database *i*.

*e*., The *Arabidopsis* Information Resource 10 or TAIR10 with respective genome version TAIR10 (https://www.*Arabidopsis*.org/) (S1 Table). The OFP sequences of *Oryza sativa* were curated from its respective genome database *i.e*., EnsemblPlants release 56 (https://plants.ensembl.org/Oryza_sativa/Info/Index) with the respective genome version 7.0 [47, 48]. The required genomes of the *Brassica* species *viz*. *B. oleracea* (v1.0), *B. nigra* (v2), *B. rapa* (v1.3), *B. carinata* (v1), *B. juncea* (v1), and *B. napus* (v1) were obtained from the *Brassica* Genomics Database or BRAD version 3.0 (http://*Brassica*db.cn/), Phytozome (https://phytozome-next.jgi.doe.gov/) and EnsemblPlants (https://plants.ensembl.org/index.html) [49, 50]. The query sequences were used to sort out and screen the OFP genes from these genomes using the $BLAST_P$ algorithm at an Expect-value threshold of $1e^{-5}$, 500 number of retained hits, 250 number of alignments, and all other parameters were kept at default [51, 52]. The $BLAST_P$ algorithm is available as the 'Several Sequences to a Big File' function in TBtools (a Toolkit for Biologists integrating various biological data-handling tools) software version 1.108 [53]. The respective OFP redundance-free domains were curated using the NCBI-CDD (National Center for Biotechnology Information–Conserved Domain Database) (an NCBI Batch Web CD-Search Tool) (https://www.ncbi.nlm.nih.gov/Structure/bwrpsb/bwrpsb.cgi) in the repurposed Pfam database (https://www.ebi.ac.uk/interpro/) as a search parameter with a Expect-value cut-off of 0.01 [54–58]. The statistical association between the ploidy level, the genome size, and the total number of identified OFP genes was calculated by determining the Pearson's Correlation coefficient value (r) [59, 60] using Minitab software version 21.3 (www.mintab.com) [61].

## 2.2 Estimation of physiochemical properties, subcellular localization prediction, and chromosomal distribution

The molecular weight (MW) and isoelectric point (pI) values of the protein sequences were calculated using the "compute pI/Mw tool" function provided by ExPASy (https://web.expasy.org/compute_pi/): a bioinformatics resource portal of the Swiss Institute of Bioinformatics (SIB) [62, 63]. The subcellular localization was predicted using the subcellular localization prediction tool BUSCA (https://busca.biocomp.unibo.it/) (Bologna Unified Subcellular Component Annotator) (S2 Table) [64]. Chromosomal localization maps were made using the 'Gene Location Visualize from GTF/GFF' function of Tbtools [51].

## 2.3 Multiple Sequence Alignment (MSA) and phylogeny

The multiple sequence alignment of all the OFP sequences was performed using the online web-based tool Clustal Omega (ClustalΩ) (https://www.ebi.ac.uk/Tools/msa/clustalo/) at default parameters [53, 65–70]. These aligned sequences were then analyzed using MEGA (Molecular Evolutionary Genetics Analysis) version 11.0.13 to assess the phylogenetic relationships [71]. For this purpose, a phylogenetic tree was constructed using the Saitou and Nei's Neighbor-Joining (NJ) Algorithm [72–80]. The phylogeny was tested using the Bootstrap method with 1000 Bootstrap reiterations for maximum statistical reliability [81–88]. Jones-Taylor-Thornton (JTT) algorithm was used as a substitution model for the respective amino acid sequences [89]. This amino acid substitution model of evolution was chosen based on the goodness of fit inferred by the Bayesian Information Criterion (BIC) and Akaike Information Criterion (AIC), both of which depend upon the maximum likelihood function (MLE) [90–94]. The rates and patterns were Gamma Distributed (G) with a shape parameter value of 5.00 [95, 96]. Gaps and the missing data were subjected to a partial deletion treatment with a site coverage cutoff percentage of 95%. The graphical representation of the phylogenetic relationship tree was made using the online phylogenetic tree annotation and display tool iTOL

(Interactive Tree Of Life) version 6.7.3 [97]. Moreover, the WebLogo 3 version 2.8.2 (https://weblogo.berkeley.edu/) web-server was utilized to develop the sequence logo illustrations for the in-depth visualization conserved domains (S1 Dataset) [98, 99].

## 2.4 Syntenic relationship analysis

The synteny and collinearity analysis was performed using the 'One Step MCScanX' function of Tbtools [100, 101]. The resultant illustrations of the syntenic analysis were made using the 'Multiple Synteny Plot' function of Tbtools.

## 2.5 Gene structure, motif distribution, promotor region analysis, and conserved domains

The 'Gene Structure View (Advanced)' function of TBtools was used to visualize the gene structure [51]. The promoter region analysis of *cis*-acting regulatory elements was performed through the web-based PlantCARE (https://bioinformatics.psb.ugent.be/webtools/plantcare/html/) utility [102]. The respective DNA sequences were given as input and the $\log_2$-normalized frequency heatmap of these *cis*-acting regulatory elements was generated through Tbtools' 'Heatmap' function (S3 Table and S1 Dataset). The lengthwise pattern of *cis*-acting regulatory elements on genes was visualized using the 'Basic Bioseqeunce View' function of Tbtools. To discover the novel, ungapped (conserved) motifs from the amino acid sequences, MEME (Multiple Em for Motif Elicitation) version 5.5.2 (https://meme-suite.org/meme/tools/meme) from MEME suite was utilized [103, 104]. All the MEME parameters were kept at default except the minimum number of motifs was notched up to 100 and the minimum width of the motif was increased to 12 in the 'advanced options' setting. Furthermore, the sequence of protein motifs curated through the MEME suite was searched for conserved domains through the aforementioned NCBI Batch Web CD-Search Tool with the same settings and parameters (S4 Table). The output conserved domains were visualized using the 'Basic Bioseqeunce View' function of Tbtools.

## 2.6 RNA-seq expression analysis

The RNA-seq expression data was retrieved through NCBI's GEO (Gene Expression Omnibus) Accession Datasets (https://www.ncbi.nlm.nih.gov/gds) [105, 106]. The high throughput evolutionary and dynamic gene expression profiling of embryogenesis and seed coat development in all members of U's Triangle was performed by Peng Gao and colleagues in 2021. The RNA-seq data from this study was used to build the relative tissue-specific expression heatmap and is available under BioProject ID: PRJNA641876 and, SRA ID: SRP268811 (S5 Table) [107]. The respective expression heatmap of relative tissue-specific expression for all members of U's Triangle was also generated through Tbtools' 'Heatmap' function. The biotic and abiotic stress RNA-seq data for *B. napus* was retrieved from *Brassica*EDB version 1.0 (https://*Brassica.*biodb.org/) [108]. On the other hand, the same data for *B. oleracea* was retrieved through NCBI's BioProject database (https://www.ncbi.nlm.nih.gov/bioproject/) available under their respective BioProject IDs *viz.* PRJNA641876, PRJNA524852, and PRJNA524852 (S6 Table) [109]. The quantification estimates and conversion of raw expression transcripts in TPM were performed using Salmon version 1.9.0+galaxy2 and Sailfish version 0.10.1.1, both web-based tools of the Galaxy server (https://usegalaxy.org/) [110–112]. Furthermore, the $\log_2$-normalized and hierarchically clustered heatmaps for *B. napus* and *B. oleracea* were developed using Morpheus (https://software.broadinstitute.org/morpheus/) [113–116].

## 2.7 Protein-protein interaction network, orthology, GO, and KEGG analysis

Protein-protein network analysis of all the protein sequences was performed using STRING-db (Search Tool for the Retrieval of Interacting Genes/Proteins) utility (https://string-db.org/) [117]. Additionally, orthologous cluster comparison analysis was performed using the web-based OrthoVenn2 utility, (https://orthovenn2.bioinfotoolkits.net/home) (S1 Dataset) [118]. The default Expect-value of $1e^{-2}$ was used as a significance threshold parameter along with the Markov Cluster Algorithm (MCL) process Inflation value of 1.5 [119]. The shared orthologous cluster representations were made according to John Venn's method of mechanical representation of propositions [120–122]. GO (Gene Ontology) term enrichment [release 05–2023] and KEGG (Kyoto Encyclopedia of Genes and Genomes) [release 106.0] pathway analysis was performed using the 'g:GOSt' suite of the g:Profiler by inputting the EnsemblPlants database compatible gene IDs at default parameters (S7 Table and S1 Dataset) [123–126]. For deeper insights into gene set enrichment, the analysis was further extended using KOBAS-i [KOBAS version 3.0] (http://bioinfo.org/kobas) [127]. The calculated enrichment score and pathways were then visualized using the SRplot web server (http://www.bioinformatics.com.cn/srplot).

## 2.8 RNA extraction and qRT-PCR expression quantification

The expression pattern of OFP genes of *B. napus* (*BnaOFPs*) under various stressors was quantified using a progressive qRT-PCR (Quantitative Real-Time Polymerase Chain Reaction) [128]. These stress conditions include salinity (150 mM NaCl), cadmium (205 mM $CdCl_2$), copper (100 mM $CuSO_4$), drought (25 g/L PEG-6000), heat (42˚C), and cold (8˚C) with a treatment duration of 2, 4, and 6 hours each. The Zhongshuang 6 (ZS6) variety of *B. napus* that was used in this experiment was provided by the Oil Crop Research Institute, Chinese Academy of Agricultural Sciences (OCRI-CAAS), Wuhan, China. A minimum of three biological and three technical replicates were assessed in each qRT-PCR experiment for general statistical reliability and robustness.

Initially, the RNA was isolated from the plant tissue samples using RNAprep Pure Plant Kit (TIANGEN, Beijing, China) according to the manufacturer's instructions. The subsequent concentration and purity of the acquired RNA samples were determined using NanoDrop™ 2000 spectrophotometers (Thermo Fisher Scientific–Waltham, Massachusetts, USA) (S8 Table). The genomic DNA contamination was eliminated using RNase-Free DNase I (TIANGEN, Beijing, China) before the cDNA synthesis. The first cDNA strand was reverse transcribed through QuantiTect Reverse Transcription Kit (QIAGEN, Hilden, Germany) using a precisely calculated amount (1000 / x–ng$\mu$L$^{-1}$ of RNA) of the isolated RNA. Finally, the SYBR™ Green qPCR Master Mix (Novogene, Beijing, China) was used to run qRT-PCR on a LightCycler Ⓡ 480 System (Roche, Basel, Switzerland) using the prepared 2$\mu$L cDNA sample. Each 20$\mu$L reaction iteration subsequently contains 10$\mu$L 2X SYBR™ Green qPCR Master Mix, 0.5$\mu$L forward primer (0.25 $\mu$M), 0.5$\mu$L reverse primer (0.25 $\mu$M), 7$\mu$L dd$H_2$O, and 2$\mu$L cDNA per PCR tube. The reaction conditions include the reverse transcription at 60˚C for 45 minutes followed by PCR amplification with initial denaturation at 95˚C for 30 seconds, denaturation at 95˚C for 15 seconds, annealing at 60˚C for 30 seconds, extension at 72˚C for 45 seconds, and final denaturation at 95˚C for 30 seconds. The main amplification reaction was repeated for a total of 40 cycles.

The transcript data through the melting curve was curated using the manufacturer-provided software suite. The *B. napus ACTIN* gene was used as an internal frame of reference and the relative normalization of gene expression was performed by implementing the standard $2^{-\Delta\Delta C_T}$ comparison method [129]. The specific forward and reverse primers for both the

*ACTIN1* and the selected *BnaOFP* genes were developed using the web-based Primer3 tool (S8 Table) [130]. The homology of each developed primer of the coding region was assessed by deploying BLAST against the *B. napus* genome to eliminate any possibility of cross-amplification during the qRT-PCR [131].

## 3. Results

### 3.1 Identification of OFP genes

A total of 55, 55, 57, 32, 28, and 29 OFP genes were identified in *B. carinata*, *B. juncea*, *B. napus*, *B. nigra*, *B. oleracea*, and *B. rapa* respectively. These sequences have been renamed linearly *i.e.*, *BcaOFP01-55*, *BjuOFP01-55*, *BnaOFP01-57*, *BniOFP01-32*, *BolOFP01-28*, and *BraOFP01-29* (S1 Table). A strong positive correlation was observed between the number of identified OFP genes and the genome size. Numerically, Pearson's Correlation coefficient value (r) was $r(4) = .9445$, $p = .004535$ with an $R^2$ value of 0.8921. In terms of assimilation of sequences from parent to progeny, the common sum expected number of sequences for *B. carinata* was 60, but only 55 non-redundant sequences were observed. Similarly, the expected number of sequences for *B. juncea* was 61, but the same as for *B. carinata*, only 55 non-redundant sequences were observed. *B. napus* was the only one that came right upon the expected value of 57 non-redundant sequences.

### 3.2 Physicochemical properties and subcellular localization

The gene length in *BcaOFPs* ranged from 212 bps (Base Pairs) up to 3345 bps with an average length of 772.58 bps. The respective protein length in *BcaOFP* ranged from 70 aa (Amino Acids) to 346 aa with an average length of 239.29 aa. In the case of *BjuOFPs*, the gene length ranged from 203 bps to 3925 bps with an average length of 883.4 bps while the protein length ranged from 68 aa to 386 aa with an average length of 243.76 aa. In the case of *BnaOFPs*, the gene length ranged from 266 bps to 2963 bps with an average length of 870.68 bps while the protein length ranged from 88 aa to 356 aa with an average length of 250.28 aa. In the case of *BniOFPs*, the gene length ranged from 230 bps to 1032 bps with an average length of 736.53 bps while the protein length ranged from 76 aa to 339 aa with an average length of 235.71 aa. Similarly, *BolOFPs* exhibited a gene length ranging from 446 bps to 2120 bps with an average length of 844.60 bps while the protein length ranged from 148 aa to 409 aa with an average length of 256.82 aa. Lastly, in *BraOFPs*, the gene length ranged from 266 bps to 2259 bps with an average length of 860.51 bps while the protein length ranged from 88 aa to 403 aa with an average length of 258.89 aa (S2 Table).

In the case of *B. carinata*, the expected lengths of both the genes and proteins hovered near the observed values. On the contrary, *B. juncea* showed a considerable deviation from this trend and its observed gene length values were slightly more than the gene length of its parent *B. rapa* but significantly higher than its other parent *B. nigra*. On the other hand, the observed protein lengths were close to the expected lengths. The reason behind the deviation in gene lengths can be attributed to possible segmental duplication during evolutionary polyploidization processes. Moreover, the gene and protein lengths in *B. napus* were also close to the expected values. The expected gene and protein lengths were calculated using the arithmetic mean of the average lengths of the sequences of the parent species.

In the physicochemical analysis, the molecular weight (MW) range of OFPs was from 8.28 kDa to 39.27 kDa in *B. carinata*, from 7.50 kDa to 43.05 kDa in *B. juncea*, from 10.31 kDa to 41.51 kDa in *B. napus*, from 8.73 kDa to 39.44 kDa in *B. nigra*, from 16.89 kDa to 45.75 kDa in *B. oleracea*, and from 10.31 kDa to 44.82 kDa in *B. rapa* (S2 Table). The range of Isoelectric point (pI) in *B. carinata* was from 4.29 to 11.31 with 37 genes being in Basic and 18 of them in

the acidic spectrum. From 4.44 to 10.41 in *B. juncea* with 34 genes in basic and 21 in the acidic range. From 4.44 to 10.74 in *B. napus* with 37 genes in basic and 20 in the acidic range. From 4.49 to 10.41 in *B. nigra* with 20 genes in basic and 12 in the acidic range. From 4.56 to 10.36 in *B. oleracea* with 18 genes in the basic and 10 in the acidic range. Lastly, from 4.56 to 10.42 in *B. rapa* with 18 genes in basic and 11 in the acidic range (S2 Table).

The analysis of subcellular localizations of OFPs revealed that in *B. carinata*, 30 genes were localized in the chloroplast (including 3 in the outer membrane), 21 in the nucleus, 3 in the mitochondrion, and 1 in extracellular space. Similarly, in the case of *B. juncea*, 30 genes were in the chloroplast (including 3 in the outer membrane), 23 in the nucleus, 1 in extracellular space, and 1 in the mitochondrion. In, *B. napus*, 32 genes were in the chloroplast (including 2 in the thylakoid and 2 in the outer membrane), 23 in the nucleus, and 2 in extracellular space. Following *B. nigra*, 16 genes in the chloroplast (including 1 in the outer membrane), 14 in the nucleus, 1 in the extracellular space, and 1 in the mitochondrion. In *B. oleracea*, 16 genes were in the chloroplast (including 3 in the outer and 1 in the thylakoid membrane), 11 in the nucleus, and 1 in extracellular space. Lastly, in *B. rapa*, 17 genes were in the chloroplast (including 1 in the outer membrane), 11 in the nucleus, and 1 in extracellular space (S2 Table).

### 3.3 Chromosomal distribution

In terms of chromosomal distribution and mapping, *BcaOFPs* were mapped on chromosomes B01-B06, B08, and C01-C09, *BjuOFPs* on A01-A03, A05-A10, B01-B03, B05, B06, and B08, *BnaOFPs* on A01-A05, A05_random, A06, A07, A09, A10, Ann_random, C01-C05, C07-C09, and Cnn_random, *BniOFPs* on B01-B03 and B05-B08, *BolOFPs* C01-C09, and *BraOFPs* A01-A10 (Fig 1). Additionally, the scaffold-inhabiting genes include *BcaOFP52-55* and *BjuOFP46-55*. *B. carinata* had a significant rearrangement of OFPs on the chromosomes in the BB-type genome and none of the OFPs were present on chromosome ChrB07. The chromosome ChrB04 in *B. carinata* has two sequences *i.e.*, *BcaOFP12* and *BcaOFP13*. Some of the pairs of genes were very closely located, such as *BcaOFP02-BcaOFP03*, *BcaOFP07-BcaOFP08*, and *BcaOFP20-BcaOFP21* on ChrB01, ChrB02, and ChrB06 respectively. On the other hand, the BB-type parent genome of *B. nigra* had four such gene pairs *i.e.*, *BniOFP01-BcaOFP02*, *BniOFP19-BcaOFP20*, *BniOFP27-BcaOFP28*, and *BniOFP31-BcaOFP32* on chromosomes B1, B6, B7, and B8 respectively. Similar to *B. carinata*, the CC-type genome from *B. oleracea* also had a significant rearrangement of OFPs but in this case, all the chromosomes were present. The CC-type in *B. carinata* had three closely related gene pairs *i.e.*, *BcaOFP28-BcaOFP29* and *BcaOFP33-BcaOFP34* both on ChrC02 and *BcaOFP44-BcaOFP45* on ChrC06. Similarly, *B. oleracea* had four such pairs which includes *BolOFP11-BolOFP12* and *BolOFP15-BolOFP16*, both on chromosome C04, *BolOFP18-BolOFP18* on C06, and *BolOFP21-BolOFP22* on C08.

In *B. juncea*, the significant rearrangement of OFPs was observed on chromosomes. For example, in the AA-type genome, none of the OFPs were present on chromosome A04 instead of the expected three OFPs present on chromosome B04 of its parent *B. rapa*. The only closely located gene pair observed is *BjuOFP17-BjuOFP18* on ChrA09. On the other hand, *B. rapa* has two such pairs *i.e.*, *BraOFP12-BraOFP13* on chromosome A04 and *BraOFP14-BraOFP15* on A05. In the BB-type genome, none of the OFPs were present on chromosome B07. The two closely located pairs are *BjuOFP31-BjuOFP32* on chromosome B08 and *BjuOFP38-BjuOFP39* on chromosome B01. On the other hand, *B. nigra* has 4 such pairs.

In *B. napus*, although the number of observed OFPs are same as expected, there are rearrangements present to a significant extent. In the AA-type genome, none of the OFPs were present on chrA08. Instead, some OFP sequences are present unevenly on random segments of different chromosomes. For example, *BnaOFP13* was present on chrA05_random while

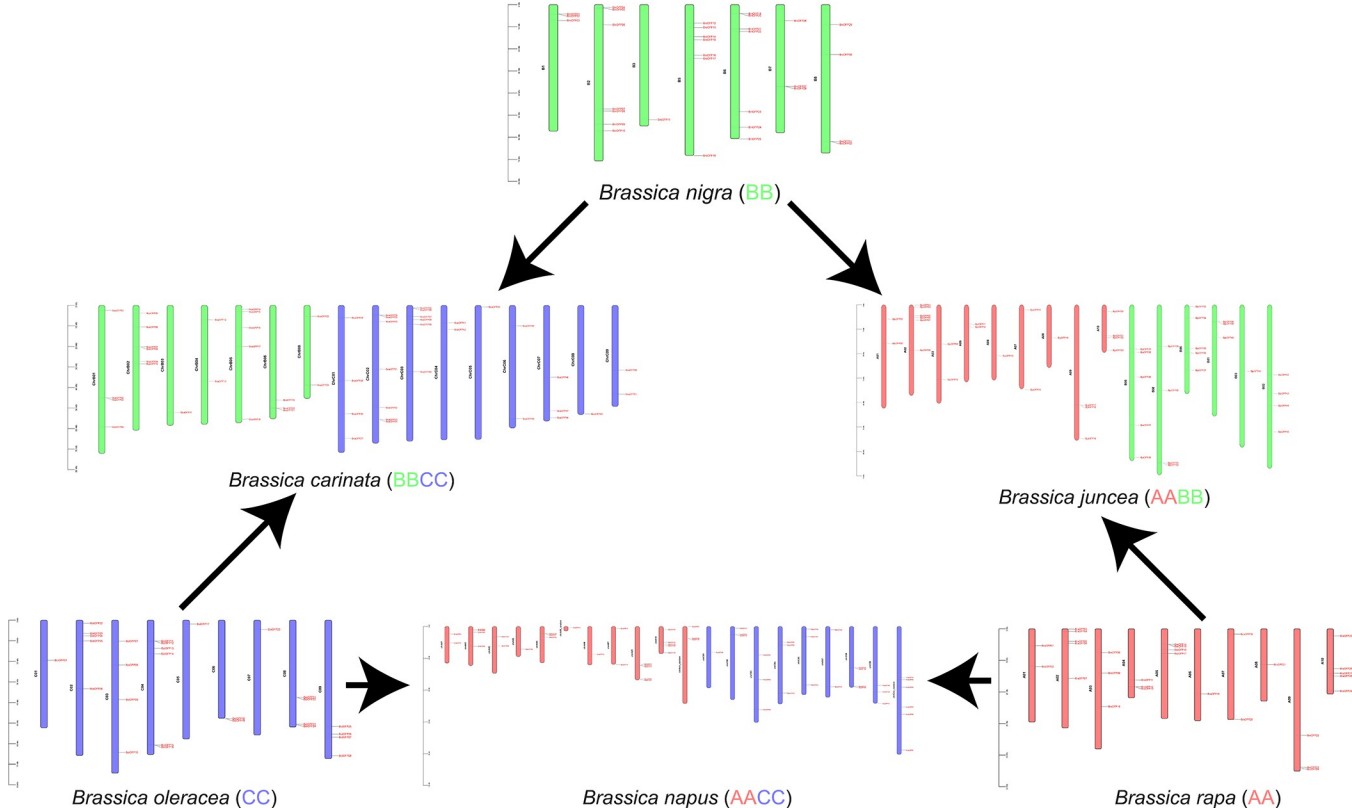

**Fig 1. Chromosomal distribution of OFP genes in the U's Triangle members.** The chromosomal distributions have been arranged following their positions in U's Triangle. The red color in the chromosomes represents the AA-type genome, green represents the BB-type, and the CC-type is represented in blue.

*BnaOFP27* and *BnaOFP28* were located on chrAnn_random. The closely related OFP gene pairs include the *BnaOFP10-BnaOFP11* on chrA05, a triad of *BnaOFP17-BnaOFP18-BnaOFP19* on chrA09 and another pair of *BnaOFP28-BnaOFP29* on chrAnn_random. *B. rapa* also shows two OFP gene pairs. In the CC-type genome, none of the OFP genes were present on chrC06. Instead, the last six sequences in the series *i.e.*, from *BnaOFP52* to *BnaOFP57* were located on chrCnn_random. The only closely located gene pair includes *BnaOFP45-BnaOFP46* on chrC08. On the other hand, *B. oleracea* shows three gene pairs.

## 3.4 Phylogenetic relationship inference

A combined phylogenetic tree of all the 256 OFPs was constructed along with the of *A. thaliana* and *O. sativa*. All of the 303 sequences have been subdivided into 4 cladded groups *i.e.*, group 1, 2, 3, and 4 as per the relative density of branches that the tree exhibited (Fig 2) (S1 Dataset). group 4 was the largest and encases 114 genes followed by group 1 holding 89, group 2 with 55, and group 3 with 45 genes. In terms of orthologous gene pairs in each clade, group 1 exhibited 7 pairs descending from *B. nigra*, 5 from *B. oleracea*, and 3 from *B. rapa* into their respective progeny species. Similarly, group 2 showed that 5, 5, and 1 orthologous gene pairs were originating from *B. nigra*, *B. oleracea*, and *B. rapa* respectively. Following the same distribution pattern, group 3 has 4, 4, and 3 orthologous gene pairs descending from *B. nigra*, *B. oleracea*, and *B. rapa* respectively. Lastly, group 4 contained 7, 3, and 4 orthologous gene pairs originating from *B. nigra*, *B. oleracea*, and *B. rapa* respectively.

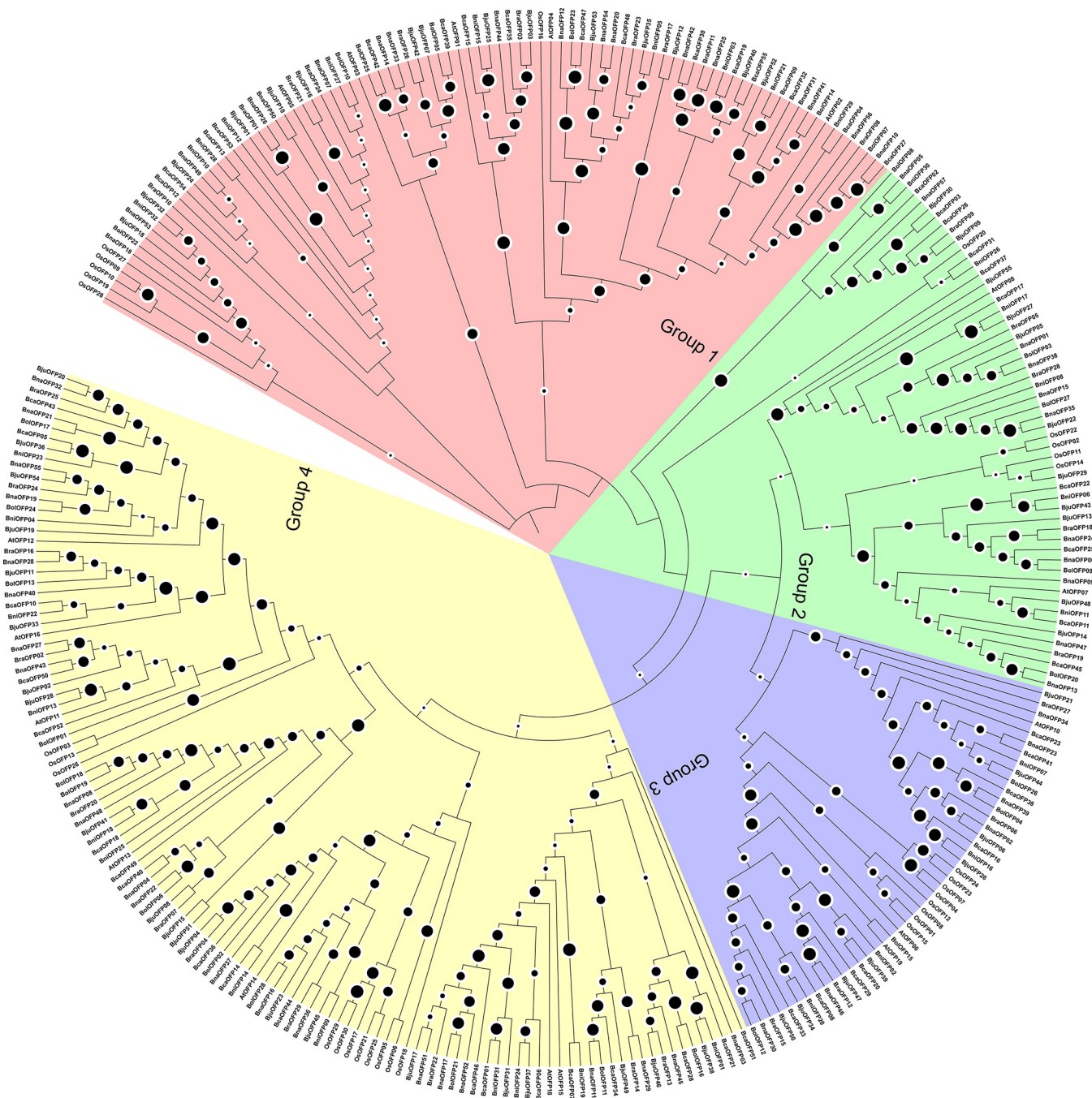

**Fig 2. Phylogenetic relationship tree of 256 OFPs of U's Triangle members along with 17 *A. thaliana* and 30 *O. sativa* OFPs.** These sequences have been subdivided into four cladded groups *i.e.*, group 1 (Red), group 2 (Green), group 3 (Blue), and group 4 (Yellow). The size of the black circles on the branches represents the relative number of Bootstrap reiterations.

In group 1, *B. carinata* had 1 duplicated gene, 4 genes were observed to be lost in *B. juncea*, and 2 duplicated genes were observed in *B. napus*. Similarly, in group 2, *B. carinata* experienced the loss of 1 gene, no genes were observed to be lost or duplicated in *B. juncea*, and only 1 duplicated gene was observed in *B. napus*. In group 3, only *B. napus* experienced a loss of 2 genes while no gene duplicates or lost genes were observed in *B. carinata* and *B. juncea*. Lastly, in group 4, *B. carinata* lost 4 genes followed by 2 lost genes in *B. juncea* and 1 lost gene in *B. napus*.

### 3.5 Multiple syntenic relationships

The gene synteny of OFPs was subdivided as per the distribution of AA, BB, and CC-type genomes (Fig 3). All these types were aligned with *A. thaliana* to better infer the evolutionary orthologous gene linkage, comparative homology, and interspecific genomic collinearity. In terms of collinear orthologous OFPs of AA-type genomes, *B. juncea* showed 8 collinear gene pairs in *A. thaliana* followed by *B. rapa* with 9 and *B. napus* with 8 pairs. In *B. juncea*, chromosome A08 exhibited a single ortholog moving from its parent *B. rapa*. *B. napus* was unable to receive this specific ortholog. Moreover, none of the OFPs followed their descent from chromosome 1 of *A. thaliana*. In BB-type genomes, *B. oleracea* presented 10 and *B. carinata* presented 11 gene pairs. In the case of *B. carinata*, chromosome ChrC08 does not seem to share any OFPs with that of its parent *B. oleracea*. Instead, the corresponding orthologs have been shifted toward ChrC07 from the adjacent chromosome C08 of *B. oleracea*. Similarly, chromosome C06 from *B. oleracea* also does not have any originating orthologs to either the *B. carinata* or *B. napus*. Lastly, in the case of CC-type genomes, *B. nigra* showed 42 collinear gene pairs in *A. thaliana*. Moreover, it can be observed that none of the orthologs originated from *B. nigra* to both *B. carinata* and *B. juncea* on chromosomes B7, B07, and ChrB07 respectively. The orthologs from chromosome B03 have been translocated to chromosome B05 from *B. nigra* to *B. juncea*. Most of the orthologs of *B. nigra* have been descending directly from *A. thaliana*. It means that these sequences may have been lost due to polyploidization events in the case of both *B. carinata* and *B. juncea*.

### 3.6 Protein-protein interaction networks

The clustered interaction network revealed that many orthologous OFPs share a significant extent of functionality with *A. thaliana* (Fig 4). The centralized clustered produced 228 edge points that are significantly more than the expected number of 42 edges chosen by the algorithm for the same number of random sequences. It strongly suggests that the OFPs are strongly connected in terms of their functionality. These proteins include *OFP01-08* and *OFP10-17*. The vicinity of the centralized cluster also holds other proteins of various correlated functions. These proteins include *BLH*1, *KNAT*5, *KNAT*7, *BLH*2, *BLH*3, *BLH*4, and *BLH*6.

### 3.7 Orthologous clusters

The orthology of the OFPs gives 39 orthologous and paralogous clusters when compared to the unit value of 1 (Fig 5). All of the 39 clusters contain at least two of the species which means there are no single-copy gene clusters present. On the whole, 17 of the clusters hold all six species out of which, four unique clusters were shown by three of the species including *B. juncea* with two (one in-group, one out-group) clusters. Similarly, *B. carinata* and *B. nigra* hold one unique cluster each. The two unique clusters in *B. juncea* point out the fact that one of the clusters may have been transferred from its parent *B. nigra*. The unique cluster of *B. carinata* also shares its homology when compared to the unique cluster of its parent *B. nigra*. Furthermore, *B. napus* was the one with the greatest number of singletons (8) followed by *B. carinata* and *B. nigra* with 4 each and, then *B. juncea* with 3. No singletons were found in both *B. oleracea* and *B. rapa*.

### 3.8 Gene structure, motif distribution, *cis*-acting regulatory elements, and conserved domains

The gene structure of OFPs revealed that 5 genes in *B. carinata*, 2 genes in *B. juncea*, 10 genes in *B. napus*, 3 genes in *B. nigra*, 2 genes in *B. oleracea*, and 3 genes in *B. rapa* had introns in their sequence. A total of 100 different types of conserved protein motifs and their domains

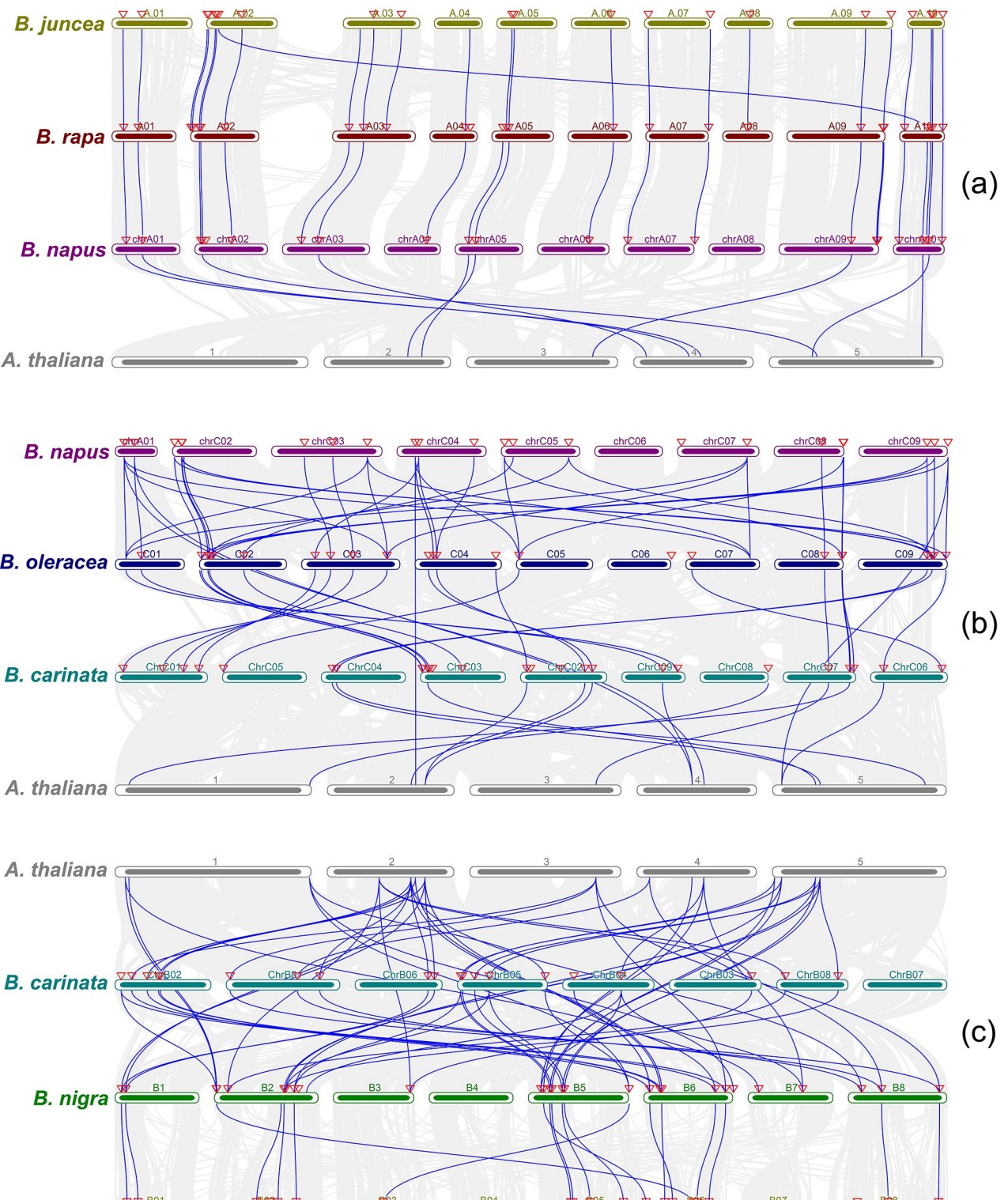

**Fig 3. Multiple syntenic relationships between the OFPs of AA, BB, and CC-type genomes and *A. thaliana*.** (a) Syntenic relationships among the OFPs of AA-type genomes and *A. thaliana*. The blue lines represent the orthologous gene pairs between genomes and the red arrow points toward the exact position of the gene on the respective chromosome. Each colored bar represents a single chromosome. The collinear syntenic blocks among genomes are represented using grey lines in the background. (b) Syntenic relationships among the OFPs of BB-type genomes and *A. thaliana*. (c) Syntenic relationships among the OFPs of CC-type genomes and *A. thaliana*.

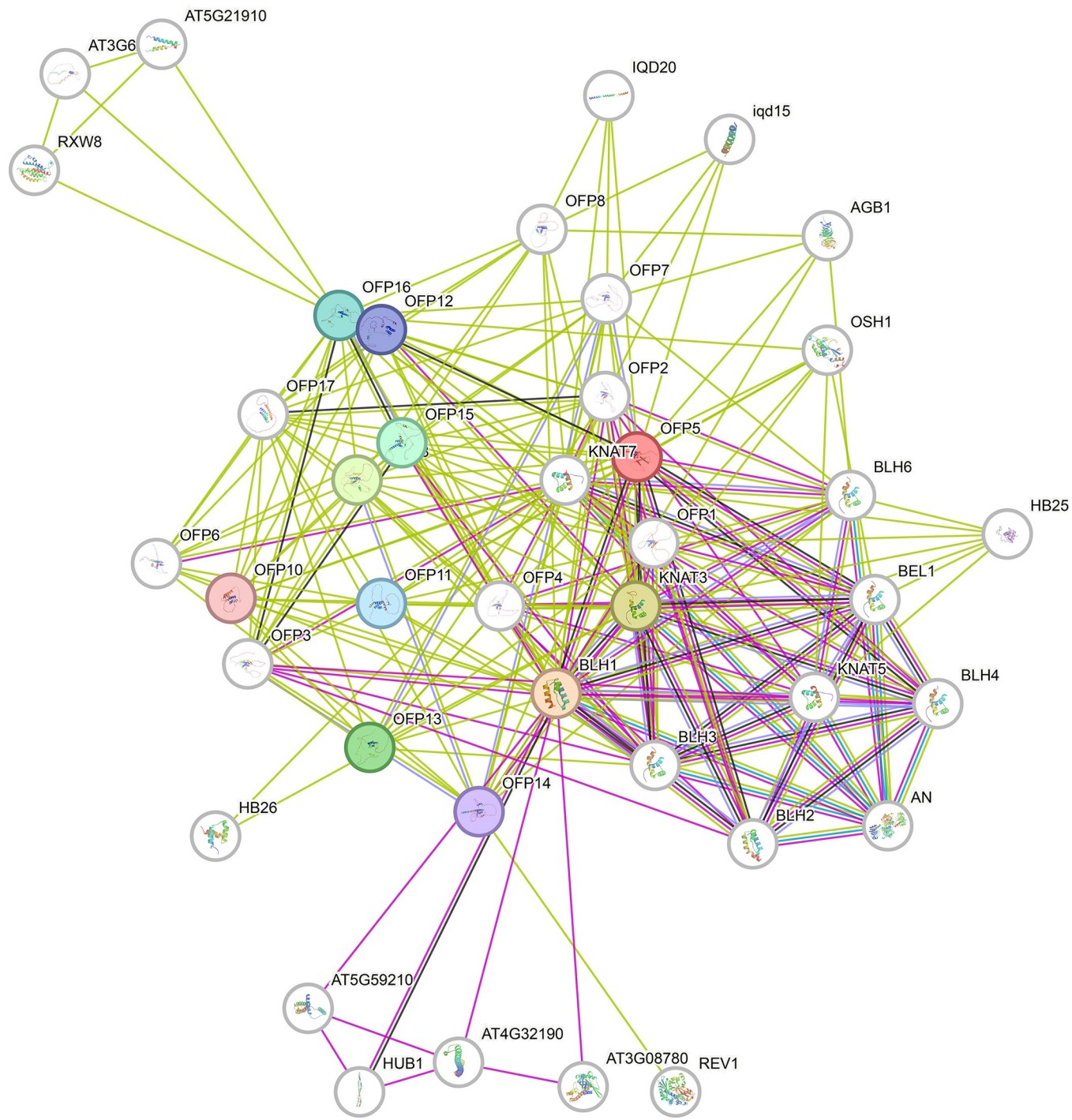

**Fig 4. Protein-protein interaction network between all OFPs and *A. thaliana*.** The globe-shape nodes represent the locus of proteins originating from a single protein-coding gene. Colored globes represent the first shell of interactions while the white globes represent the second shell of interactions. The purple lines represent the known and experimentally determined interactions while the blue lines represent interactions curated from databases. Yellow lines show the interactions curated through text-mining.

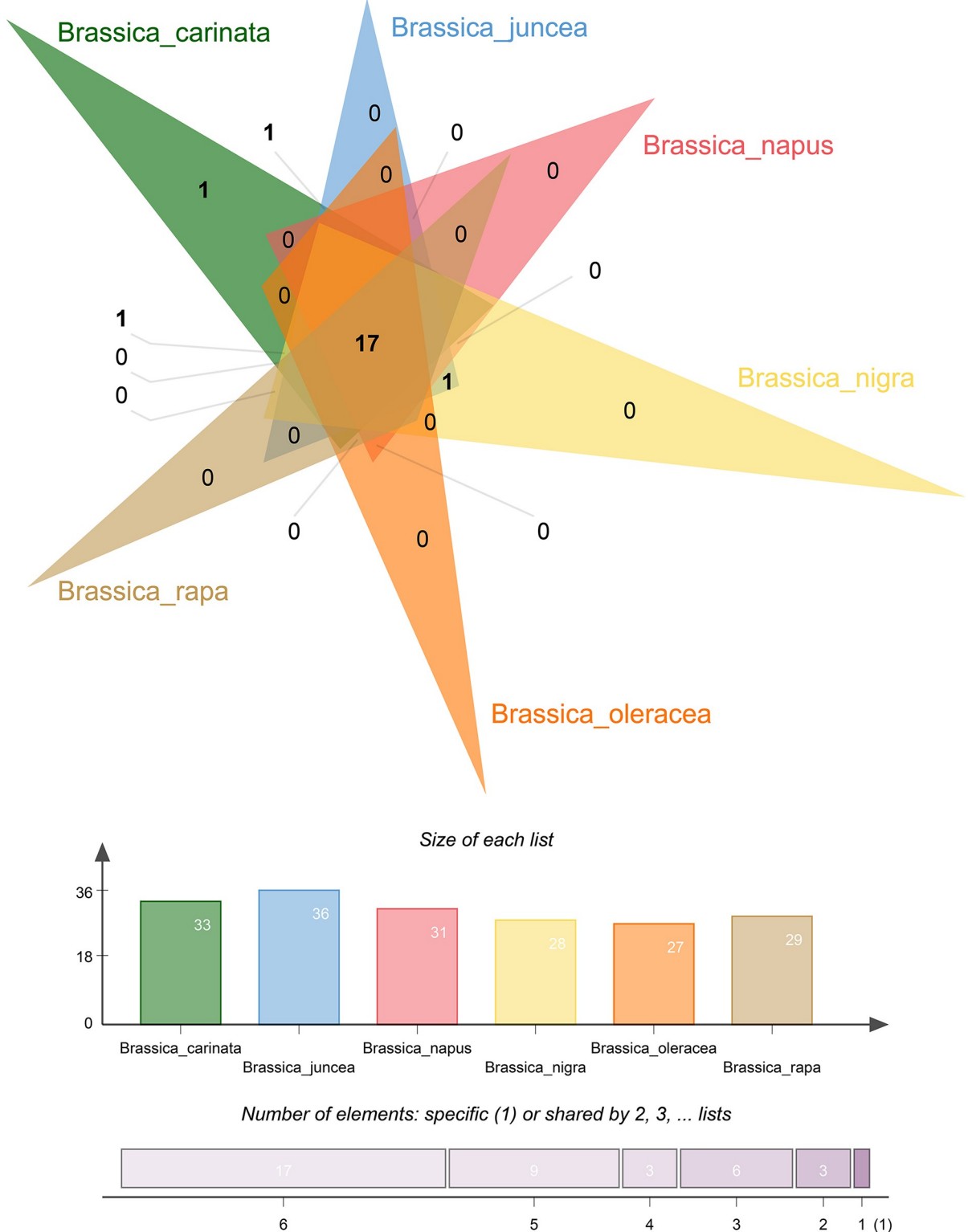

**Fig 5. Orthology and the distribution of orthologous OFP clusters.** The graph on the bottom shows the number of total individual and shared clusters by each *Brassica* species.

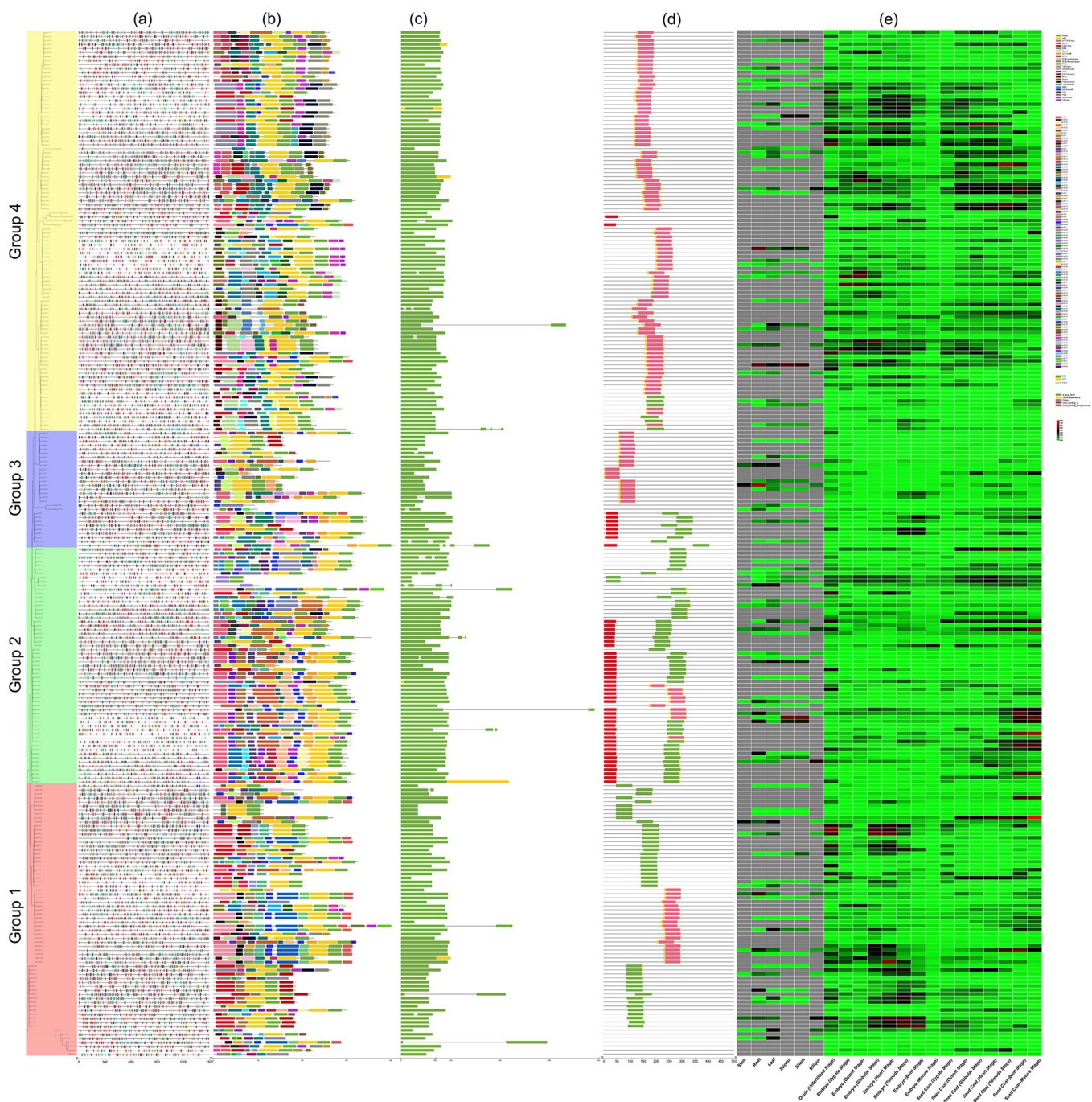

**Fig 6. Phylogenetic distributions of *cis*-acting regulatory elements, conserved protein motifs, gene structure, motif domains, heatmap of *cis*-acting regulatory elements, and heatmap of relative tissue-specific expression RNA-seq datasets.** (a) Distribution pattern of 26 *cis*-acting regulatory elements (b) 100 different conserved motifs (c) Gene structure. Scale-line on the bottom x-axis shows the length of genes in base pair (bp) units (d) Conserved motif domains (e) Relative tissue-specific expression heatmap. All the grey-colored blocks represent the absence of expression data.

were identified. The sequences that have been cladded together are clustered in terms of phylogenetic similarity combined with motif distribution (Fig 6). In the clade of group 4, motif number 4 shows the most abundance while motif numbers 3 and 16 were present in abundance in group 2. Similarly, motifs 5, 55, and 76 were dominant in group 1, and motif number

33 was dominant in group 3. Moreover, most of the sequences show a constant distribution of motifs 1 and 2.

The 1500 bp upstream distribution pattern of *cis*-acting regulatory elements in OFPs was analyzed. Prominently identified *cis*-acting regulatory elements include *CAAT-box*, *TATA-box*, *G-box*, *MRE*, *GT1-motif*, *GATA-motif*, *Box 4*, *ARE*, *ABRE*, *LTR*, *TGA-element*, *CGTCA-motif*, *TGACG-motif*, *P-box*, *WUN-motif*, and *CAT-box*. The elements of unknown function include *AT~TATA-box*, *Myb-binding site*, *ERE*, *TCA*, *MYB*, *MYC*, *AAGAA-motif*, *MYB-like sequence*, *TATA*, and *as-1*. *Box 4* was the most abundant element followed by *ARE*, *G-box*, and *ABRE*. The frequency heatmap of all these *cis*-acting regulatory elements shows the uniform frequency of all these elements (Fig 6). Other mildly expressed *cis*-acting regulatory elements include *AT~TATA-box*, *MYB*, *MYC*, *ERE*, *CGTCA-motif*, *TGACG-motif*, and *as-1*.

The conserved domains of OFPs were also characterized and mapped (Fig 6). Following the phylogenetic clade distribution, group 4 exhibited the most widespread distribution of *ovate* and *ovate superfamily* domains in almost all the sequences. In *BolOFP18* and *BolOFP19*, *ovate* domains were falling under the range of *3678* domain of *A. thaliana*. The *A. thaliana 3678* domain was also observed in six other sequences. *BnaOFP03* and *BjuOFP29* only displayed both *DNA binding 2* and *DNA binding 2 superfamily* domains. Moreover, three OFPs displayed no domain at all. In group 3, *ovate* and *ovate superfamily* domains remained dominant but almost one-third of the sequences displayed all three aforementioned domains at once *i.e.*, *DNA binding 2*, *DNA binding 2 superfamily*, and *A. thaliana 3678* domain. Two of the sequences display no conserved domains in this clade.

In group 2, all possible combinations of domains were observed. For instance, ten sequences displayed four conserved domains simultaneously *i.e.*, *ovate* superfamily, *DNA binding 2*, *DNA binding 2 superfamily*, and *A. thaliana 3678*. Nine other sequences displayed *ovate*, *ovate superfamily*, *DNA binding 2*, and *DNA binding 2* conserved domains. Six sequences were observed with *A. thaliana 3678* masked by *ovate superfamily* domains. Twenty sequences exhibited three domains of group 4 *i.e.*, *DNA binding 2*, *DNA binding 2 superfamily*, and *A. thaliana 3678*. Moreover, ten sequences only displayed the *A. thaliana 3678* domain. *BniOFP26* was the only sequence with no conserved domain at all. Group 1 exhibited the most linear fashion of distribution of domains. For instance, nineteen sequences displayed *ovate* and *ovate superfamily* while three others displayed both *ovate superfamily* and *A. thaliana 3678* domain. Thirty-nine sequences were observed with just *A. thaliana 3678* domain. Moreover, seven sequences were observed with no domain at all.

### 3.9 RNA-seq expression analysis

In terms of plant tissues, it can be observed that the highest upregulated gene expression values at the ovule (unfertilized stage) were exhibited by *BjuOFP40*, *BolOFP16*, *BraOFP13*, *BnaOFP28*, *BnaOFP50*, *BcaOFP15*, *BolOFP27*, *BraOFP28*, *BniOFP08*, and *BolOFP03*. Following the course of development during embryogenesis, in the embryo (zygote stage) the prominent expression values were shown by *BniOFP25*, *BraOFP20*, and *BraOFP16*. Similarly, in embryo (octant stage) *BnaOFP37*, *BcaOFP49*, *BniOFP25*, and *BraOFP20* were prominently expressed. In the embryo (globular stage), *BniOFP01*, *BolOFP13*, *BraOFP28*, *BniOFP08*, *BraOFP19*, and *BniOFP20* were significantly expressed. Similarly, in the embryo (heart stage), *BniOFP01*, *BolOFP27*, *BraOFP28*, *BniOFP08*, *BolOFP20*, and *BniOFP20* were significant. In the embryo (torpedo stage), *BjuOFP55*, *BcaOFP08*, *BniOFP20*, *BolOFP12*, and *BraOFP15* were prominent. In the embryo (bent stage), only *BniOFP20* and *BraOFP15* were expressed to a significant extent. In the embryo (mature stage), the only sequence that was expressed to a moderate extent was *BraOFP03*.

In the seed coat (zygote stage), *BolOFP16*, *BnaOFP13*, *BnaOFP28*, and *BcaOFP15* were moderately expressed. In the seed coat (octant stage), only *BolOFP02* and *BjuOFP30* were significantly expressed. In the seed coat (globular stage), only *BniOFP01* showed a considerable expression. In seed coat (heart stage), *BniOFP01*, *BraOFP29*, *BjuOFP16*, and *BjuOFP30*. In the seed coat (torpedo stage), *BraOFP04*, *BolOFP28*, *BjuOFP16*, *BniOFP15*, *BjuOFP30*, and *BniOFP11* were significantly expressed. In case of seed coat (bent stage), *BraOFP04*, *BniOFP09*, *BjuOFP16*, *BniOFP21*, *BraOFP17*, *BnaOFP12*, *BjuOFP53*, *BjuOFP09*, and *BniOFP11* exhibited relatively high expression values. Lastly, in seed coat (mature stage), *BcaOFP15*, *BniOFP15*, *BjuOFP52*, *BcaOFP09*, *BraOFP17*, *BnaOFP12*, *BjuOFP53*, *BjuOFP09*, and especially *BjuOFP30* were expressed relatively very high.

In the stem portion, *BniOFP07* and *BniOFP30* were moderately expressed. *BcaOFP40*, *BjuOFP11*, and *BniOFP07* were significantly expressed in the root portion. *BcaOFP52*, *BcaOFP10*, *BjuOFP16*, *BniOFP26*, *BcaOFP30*, *BniOFP30*, *BcaOFP11*, *BniOFP27*, and *BniOFP32* were moderately expressed in the leaf portion. In the stigma and the shoot portion, only *BcaOFP10* and *BcaOFP09* exhibited a mild expression. Lastly, *BniOFP09*, *BniOFP05*, and *BniOFP20* showed moderate expression in the silique portion.

The expression response patterns of OFPs were also evaluated in various tissues under different biotic and abiotic stressors (Fig 7A, 7B). In *B. napus*, from the observation of statistically correlated clusters, *BnaOFP29*, and its neighboring members were upregulated in the embryo at greenhouse conditions. Similarly, *BnaOFP12* and its clustering members were upregulated in endosperm tissues in the field conditions. *BnaOFP16* showed high expression values in seed tissue in both the control and test groups. *BnaOFP23* and *BnaOFP49* were also upregulated in the same test group. *BnaOFP26* and *BnaOFP51* showed very high expression values in seed tissues at the transgene-negative maturation stage and field conditions. In both, the test and control groups of seed coat tissues, the cluster *BnaOFP20*, *BnaOFP56*, *BnaOFP26*, and *BnaOFP21* exhibited high expression values. Similarly, in the control group of petals, *BnaOFP36*, *BnaOFP01*, *BnaOFP44*, and *BnaOFP28*, along with *BnaOFP13* were highly upregulated.

In bud, under SX-1 chemical stress, *BnaOFP31* showed an upregulated expression. Under the abiotic stress of chemicals like EUE, NaCl, PEG1, PEG2, and PEG-6000 in the leaf, the cluster of *BnaOFP13*, *BnaOFP37*, *BnaOFP10*, and *BnaOFP55* exhibited high expression values. In the centralized cluster of leaves at both sterile and fertile conditions, high expression values were observed in more than half of the *BnaOFP* genes. *BnaOFP24* and *BnaOFP42* were upregulated in the shoot apexes control group. The next major cluster was observed in roots under the abiotic chemical stress of NaCl, NaCl, and melatonin, and at low boron concentrations. The genes in this major cluster include *BnaOFP02*, *BnaOFP27*, *BnaOFP09*, *BnaOFP04*, *BnaOFP14*, *BnaOFP35*, and *BnaOFP57*. Among all the genes, *BnaOFP27* was observed to exhibit a consistent expression at a low temperature of 4˚C as well as in roots under various chemical stresses.

In *B. oleracea*, *BolOFP16* and *BolOFP16* were observed to be upregulated in the embryo on day 1 at 16˚C temperature ambient temperature. Similarly, at 26˚C on day 1 and day 3, the expression *BolOFP16* was still high. *BolOFP13* and *BolOFP17* showed high expression values at 26˚C on day 7. In endosperm, on days 1, 3, 7, 15, 18, and 22 at 16˚C, the cluster of *BolOFP25*, *BolOFP23*, and *BolOFP28* displayed consistent expression values. This same cluster also exhibited a consistent expression at 26˚C on days 1, 3, and 7 as well. On day 22, at the same temperature, *BolOFP18* and *BolOFP19* were upregulated with *BolOFP24* and *BolOFP22* on the 26˚C-recovery cycle on days 18 and 22 respectively. The main centralized cluster of the heatmap holds more than half of the *BolOFP* genes all expressing under the biotic stress from *Xanthomonas campestris* (Xcc) on leaf tissue in both the control and inoculation groups on days 3 and 12. Lastly, in roots, *BolOFP11* was upregulated at low phosphate and zinc concentrations. At

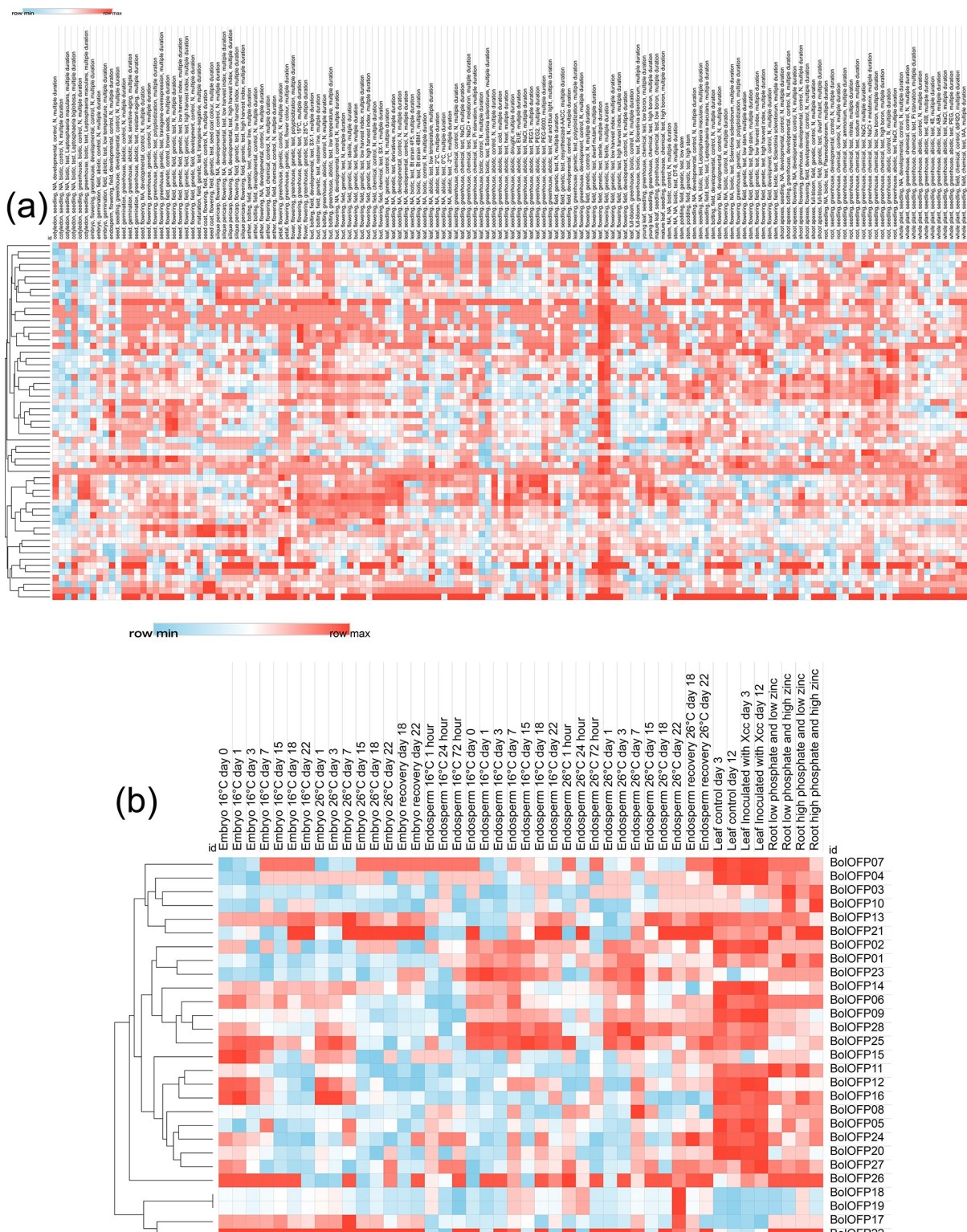

**Fig 7. Expression heatmaps of biotic and abiotic stresses for *B. napus* and *B. oleracea*.** (a) Expression heatmap of RNA-seq biotic and abiotic stress data for the identified OFP genes in *B. napus*. (b) Expression heatmap of RNA-seq biotic and abiotic stress data for the identified OFP genes *B. oleracea*.

low phosphate but high zinc concentrations, both *BolOFP01* and *BolOFP03* displayed high expression values. On the contrary, *BolOFP11* was upregulated at high phosphate but low zinc concentrations along with *BolOFP10* being highly upregulated at high phosphate and zinc concentrations simultaneously.

### 3.10 GO terms functional enrichment and KEGG pathway analysis

In *B. juncea*, it was found that 53 GO terms were found to be above the threshold $-\log_{10}(p_{adj})$ value of $\geq 16$ and therefore are significantly enriched in terms of their involvement in the regulatory Biological Processes (BPs) such as negative regulation of nucleic acid-templated transcription (Fig 8A). In the Molecular Functions (MFs) such as DNA binding, only 4 GO terms were somewhat enriched but not significant. BP also exhibited 5 non-significantly enriched GO terms. In *B. napus*, it was observed that a total of 52 GO terms were above the threshold level of significance, out of which 51 were significant for BPs like negative regulation of nucleic acid-templated transcription and 3 were significant for Chemical Components (CCs) such as in the nucleus (Fig 8B). In terms of non-significant GO terms, MFs exhibited 4 in DNA binding function, BPs exhibited 7, and 3 were exhibited by CCs.

*B. oleracea* shows the same trend as that of *B. napus* with a total of 48 GO terms above the threshold level of significance (Fig 8C). In BPs, 50 terms were found to be significantly enriched while 8 terms were observed to be non-significant. Similarly, in CCs, 3 termed were enriched and 3 were non-significant, same as before. Only 2 non-significant terms were observed in the case of MFs. Lastly, *B. rapa* follows the same trend as the previously mentioned *B. juncea* with 49 enriched GO terms in the case of BPs and 9 non-significant terms (Fig 8D). MFs showed just two non-significant GO terms and no terms were observed in the case of CCs. The detailed KEGG pathway description of these results showed other numerous functions of minor significance, in which the homologs of *BjuOFPs*, *BnaOFPs*, *BolOFPs*, and *BraOFPs* are involved as well as the interconnected pathways to the abovementioned MF, BP, and CC processes (S7 Table).

### 3.11 qRT-PCR expression pattern of OFPs in *Brassica napus*

The qRT-PCR was deployed to check the expression pattern of selected *BnaOFP* genes (12, 13, 16, 26–29, 31, 37, 50, and 51) under various stress conditions. These stress conditions include salinity, cadmium, copper, drought, heat, and cold with a treatment duration of 2, 4, and 6 hours each (Fig 9). In terms of prominence, a constant stream of upregulated expression in the case of Heat (4h) was observed in *BnaOFP12*, *BnaOFP13*, *BnaOFP16*, *BnaOFP26*, *BnaOFP28*, *BnaOFP31*, *BnaOFP50*, and *BnaOFP51*. Under salinity (6h) and cadmium (2h) treatment, *BnaOFP27*, *BnaOFP31*, and *BnaOFP37* exhibited a highly elevated expression. Especially, the *BnaOFP27* and *BnaOFP37* genes had a constant upregulated expression in the case of cadmium treatment under 2h, 4h, and 6h. The elevated expression of these two genes was also observed under the copper (2h) treatment. *BnaOFP29* had a mildly elevated expression under salinity (2h), cadmium (2h), and copper (4h) treatments. Both the *BnaOFP50* and *BnaOFP51* also had elevated levels of expression under salinity (6h) and cadmium (2h) treatments. Moreover, *BnaOFP26*, *BnaOFP27*, and *BnaOFP37* also exhibited an upregulated expression pattern under cold (6h) treatment. On the contrary, almost all the *BnaOFP* genes were highly down-regulated under all drought (2h), (4h), and (6h) treatments. A mild down-regulation was also observed in the case of cold (2h) and (6h) treatments in almost all genes followed by copper (6h) and drought (6h) in some cases (S8 Table).

## 4. Discussion

Ovate Family Proteins (OFPs) consist of a C-terminus *OVATE* domain that is approximately 70 amino acids in length. OFPs generally serve as transcription factors specific to plants and

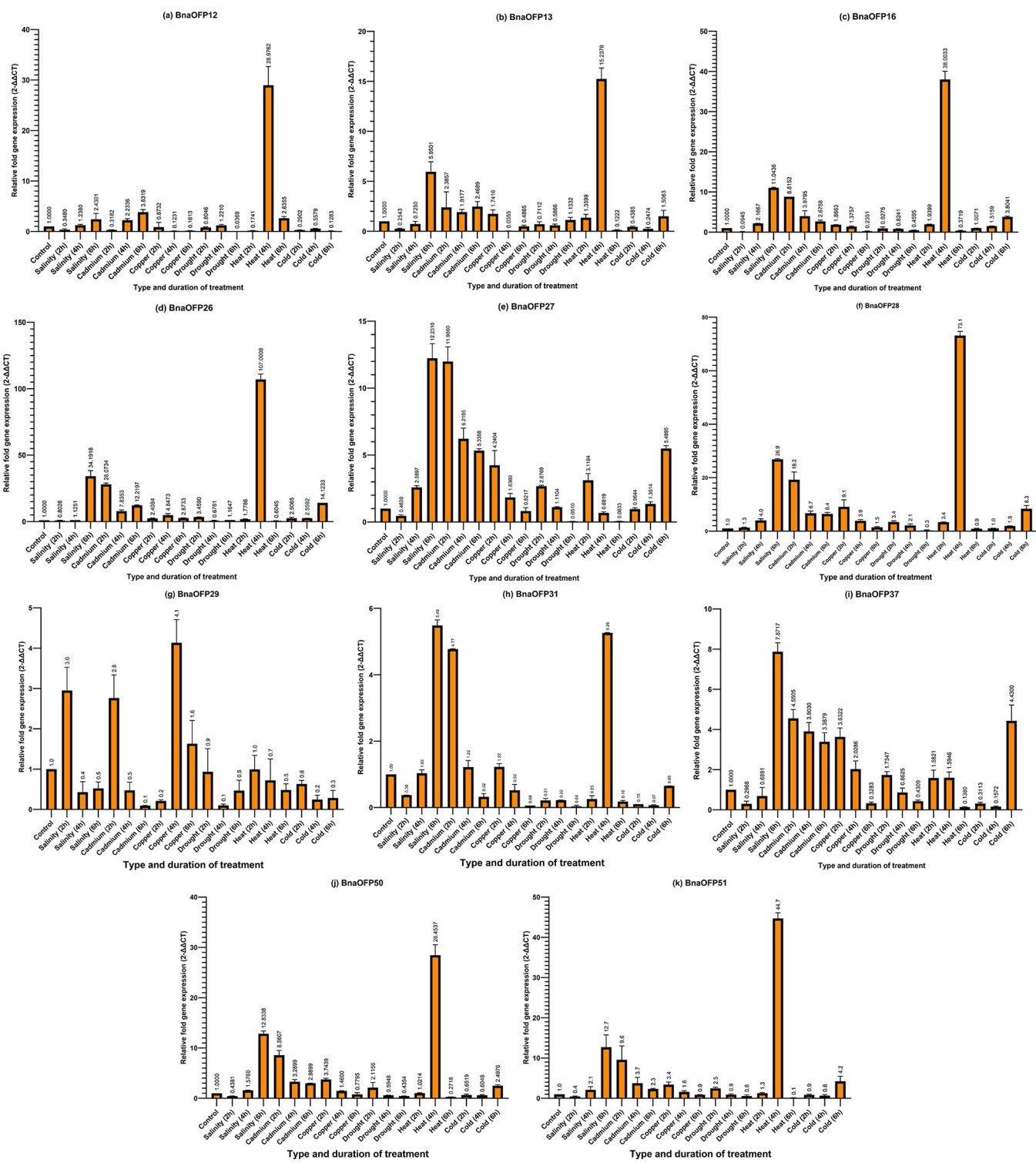

**Fig 8. qRT-PCR expression pattern of the selected *BnaOFP* genes.** In each case, the relative expression values have been plotted on the Y-axis. The X-axis represents the type and duration of the subsequent stress condition. (a) *BnaOFP12* (b) *BnaOFP13* (c) *BnaOFP16* (d) *BnaOFP26* (e) *BnaOFP27* (f) *BnaOFP28* (g) *BnaOFP29* (h) *BnaOFP31* (i) *BnaOFP37* (j) *BnaOFP50* (k) *BnaOFP51*.

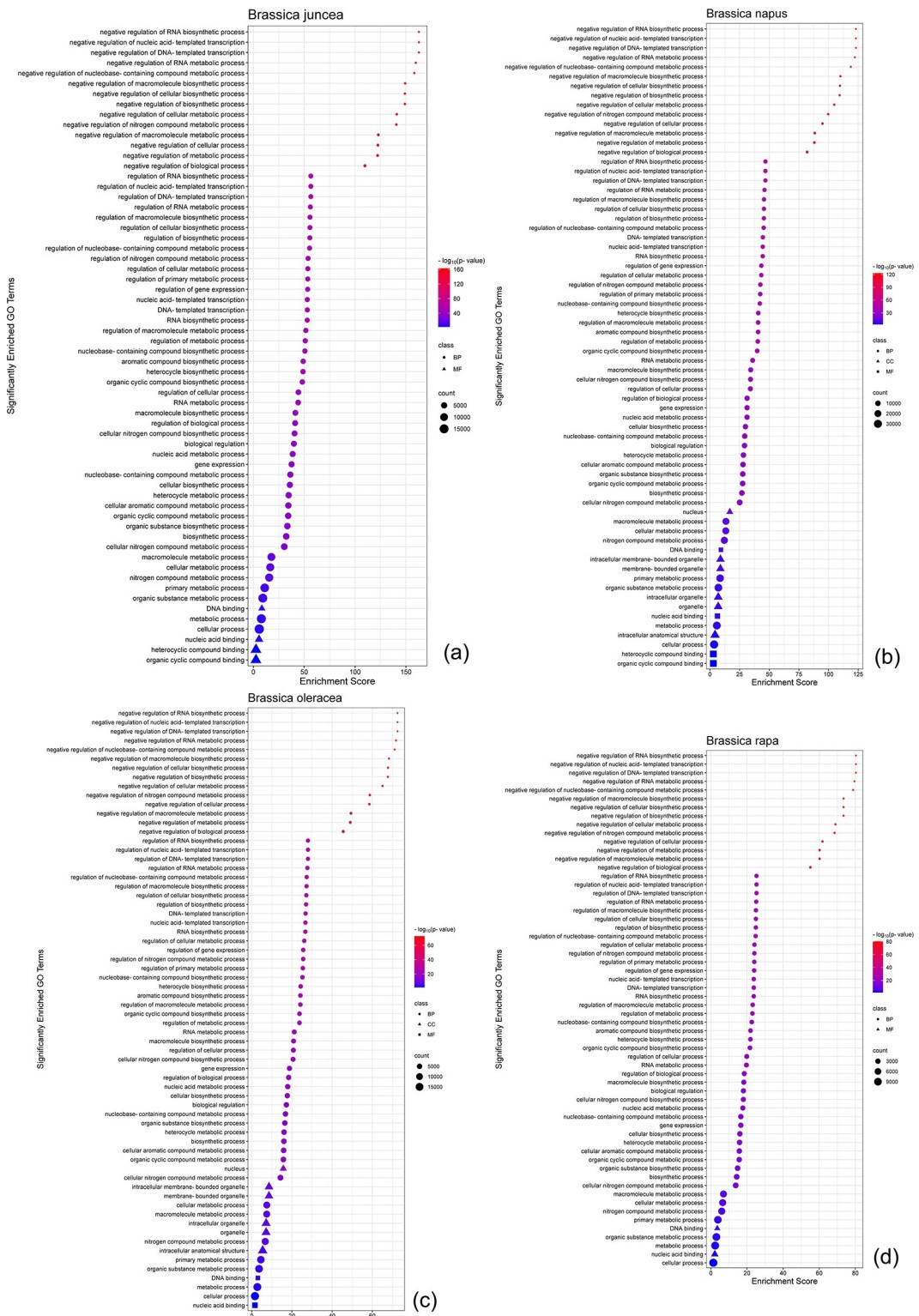

**Fig 9. Enrichment graph of significant GO terms, gene count, and distribution.** (a) *B. juncea* (b) *B. napus* (c) *B. oleracea* (d) *B. rapa.*

are localized in the nuclear region. Numerous studies have reported their presence in many plant species and most of them are diverse in nature on phylogenetic scales. They have been reported most recently in *B. rapa* [46], followed by the modal plant, *A. thaliana* [8], in radish [9], apple [132], tomato [133], and many other species [17, 19, 21, 134–136]. OFPs were initially characterized to have a role in transcriptional repression but later it was revealed that they also have diverse and complex supporting roles in conjunction with phytohormones in numerous plant growth and development processes [137, 138]. In terms of chromosomal localization of OFP genes, some genes were observed to be deficient after WGD in the progeny species and the reason behind these can be attributed to evolutionary polyploidization processes. These duplication events may have led to functional divergence, and given rise to novel uncharacterized domains and structures. For instance, thirteen OFPs were found to have no conserved domain which suggests that this may have happened during the evolutionary polyploidization that resulted in the subsequent loss of function. Although, most of the homologs of parent species were acquiring somewhat the same place in the genome of the progeny, yet some genes were also observed to be entirely relocated to other regions of the genome. Similar results have been previously reported in a study on the OFP homologs of various land plants [134]. Overall, a high degree of positive correlation between the number of identified OFPs and the size of the genomes indicated the extent of duplication process of subsequent parent genomes. Moreover, the size of the genes and proteins was consistent across all the species pointing toward the fact that when moving from parent-to-progeny, there was not a significant accumulation of mutations other than the substitutions which have not been detected as of yet. OFPs also fall in the basic pH spectrum which greatly affects their cellular solubility due to a subsequent increase in hydrophobicity.

Almost, all of the OFP proteins are localized in the chloroplast and nucleus. It can be speculated that they may have crucial roles in photosynthetic processes such as in light absorption also due to the presence of light-responsive *cis*-acting regulatory elements in the promoter region of OFPs [6, 46, 139]. The localization of OFP proteins in the nucleus points towards the previously researched activation of OFP-mediated transcriptional repression processes under biotic and abiotic stress treatments through the presence of low-temperature responsive *cis*-acting regulatory elements [140]. These transcriptional repression processes could also have been subsequently mediated by *BLH* and *KNAT* transcription factors under stress [8, 11, 141–143]. The active translational nature of the OFPs was also observed because most of the OFP genes lacked intron in their sequences [132]. The dominant presence of *TATA-box* and *CAAT-box* and other *cis*-acting regulatory elements of uncharacterized function indicated their crucial supporting roles in growth and development mechanisms [38]. Furthermore, *DNA binding 2* and its superfamily were the dominant domains to be observed and they have their reported role in the regulation of gene expression [144]. The distribution of conserved motifs also follows the phylogenetic cladded groups with motifs 1 and 2 being the most prominent in almost all the OFPs. Both of the motifs are also a part of *DNA binding 2* and its *superfamily* domain [145]. Further evidence of the involvement of OFPs in growth and development processes comes from GO and KEGG analysis. All four species that were observed for the GO and KEGG analysis showed a higher relative degree of involvement in Biological Processes (BPs) such as negative regulation of nucleic acid-templated transcription, Molecular Functions (MFs) such as DNA binding, and parts of Cellular Components (CCs) such as the nucleus. The KEGG pathway analysis shows the subsequent pathways that lead to these functions defined by the statistically significant occurrence of enriched GO terms.

The syntenic relationship analysis shows the movement and distribution of OFPs from one species to another justifying their consistency except for some previously mentioned OFPs that failed to duplicate and deleted altogether leading to some loss of associated functions [134,

135]. *OFP05* and its orthologs were found to be the main contributing factors in transcriptional repression. The interaction of *OFP05* with other transcription factors like *KNAT* and *BLH* was found to mediate several aspects of embryonic development [141]. Furthermore, its correlated functions were also observed, like with *KNAT3* in embryo sac development [11]. These overall results suggest the crucial role of OFPs in plant growth and development processes.

In terms of expression pattern in the development of the seed coat, the *BjuOFP30* gene was characterized in the tissue-specific expression analysis for its relatively higher expression in the seed coat at the mature stage. Similarly, genes such as *BnaOFP27*, *BolOFP11*, and *BolOFP10* were identified to be highly upregulated in NaCl and NaCl with melatonin treatment, in roots under low phosphate and low zinc, and high phosphate and high zinc concentrations respectively. These results suggest the role of these OFPs in the regulation of growth and development processes, and plant regulation under various biotic and abiotic stress conditions [132, 140]. The experimental validation of expression pattern through qRT-PCR confirms that the aforementioned *BnaOFP27* is highly upregulated under salinity (6h) treatment along with other stressors such as cold (6h), copper (2h), and cadmium (2h), (4h), and (6h) treatment. Further functional studies on this gene might help in regulating plant growth and development under various stress conditions to boost productivity and yield.

Combined with the domain, conserved motif, frequency pattern in *cis*-acting regulatory elements, tissue-specific, and the environmental stress expression pattern under qRT-PCR, the basis required for functionally diverse analysis of the OFP gene family could be established in the future. Moreover, the availability of more RNA-seq biotic and abiotic stress data can be used to provide more evidence for the regulatory and supportive roles of OFPs in plant growth and development processes. With a better understanding of the underlying OFP regulation mechanisms, crop improvement programs can be modified more efficiently.

## 5. Conclusions

In this study, we have isolated 256 total OFP genes from all six members of U's Triangle. The diploid and tetraploid ploidy levels in the genomes enabled us to investigate all types of correlations across the species and their sub-genomes. In terms of genome size and their respective OFP genes, a strong positive association was observed. Moreover, through protein interaction network analysis, we have found that the OFPs, other than the regular transcriptional repression, were also involved in regulating several biological processes, molecular binding functions, multiple organelles, and cellular components. Prevalence of *cis*-acting regulatory element showed the responsiveness of OFPs toward the stimulus of light, hormones, and low-temperature along with their role in regulating the expression of meristem and their reflex toward wounds. The domain and orthology analysis displayed the distribution of *ovate*, *ovate superfamily*, and other related domains that quantify the loss of function and the appearance of novel gene structures.

Tissue-specific expression analysis with genes like *BjuOFP30* further sheds light on their role in different stages of plant growth and development. Expression analysis of biotic and abiotic stresses in different tissues gave evidence to establish the supportive and regulatory roles of OFPs such as *BnaOFP27* (validated through qRT-PCR), *BolOFP11*, and *BolOFP10*. Syntenic analysis displays the evolutionary displacement and translocations of OFP orthologs during WGD events. The GO term enrichment and KEGG functional pathway further solidified the involvement of OFPs in the regulation of plant growth and development processes. Our study provides the necessary basis required as a pre-requisite for further complex and functionally diverse analysis of the OFP gene family in the future.

## Supporting information

**S1 Table. Retrieved OFP sequences.**
(XLSX)

**S2 Table. Physicochemical information of OFPs.**
(XLSX)

**S3 Table. Frequency values of *cis*-acting regulatory elements in OFPs.**
(XLSX)

**S4 Table. Information about 100 identified protein motifs in OFPs.**
(XLSX)

**S5 Table. Retrieved expression values of OFPs in different tissues and development stages.**
(XLSX)

**S6 Table. Retrieved expression values of *B. napus* and *B. oleracea* in different tissues and development stages under various biotic and abiotic stress parameters.**
(XLSX)

**S7 Table. GO terms and functional enrichment analysis datasets.**
(XLSX)

**S8 Table. Developed primers of *Brassica napus* for qRT-PCR, the calculated 2-ΔΔCT values, and the RNA quantification profile.**
(XLSX)

**S1 Dataset. Detailed illustrations of *cis*-acting reglatory elements' frequency, GO and KEGG functionally enriched pathways, evolutionary orthology, individual phylogeny, and sequence logos of the prominent OFP domains.**
(ZIP)

## Acknowledgments

The infrastructural support from the Centre for Advanced Studies in Agriculture and Food Security (CAS-AFS), at the University of Agriculture Faisalabad, Faisalabad 38000, Punjab, Pakistan is gratefully acknowledged.

## Author Contributions

**Conceptualization:** Muhammad Shahzaib, Uzair Muhammad Khan, Iqrar Ahmad Rana.

**Data curation:** Muhammad Shahzaib, Uzair Muhammad Khan.

**Formal analysis:** Muhammad Shahzaib.

**Funding acquisition:** Iqrar Ahmad Rana.

**Investigation:** Muhammad Shahzaib.

**Methodology:** Muhammad Shahzaib, Uzair Muhammad Khan.

**Project administration:** Sultan Habibullah Khan, Iqrar Ahmad Rana.

**Resources:** Muhammad Shahzaib, Sultan Habibullah Khan, Qamar U. Zaman, Iqrar Ahmad Rana.

**Software:** Muhammad Shahzaib.

**Supervision:** Muhammad Tehseen Azhar, Rana Muhammad Atif, Sultan Habibullah Khan, Iqrar Ahmad Rana.

**Validation:** Muhammad Shahzaib, Qamar U. Zaman.

**Visualization:** Muhammad Shahzaib.

**Writing – original draft:** Muhammad Shahzaib.

**Writing – review & editing:** Muhammad Shahzaib, Muhammad Tehseen Azhar, Rana Muhammad Atif.

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
