## [Decision Letter · Decision Letter 0]

29 Nov 2023

PONE-D-23-33855Phylogenomic curation of Ovate Family Proteins (OFPs) in the members of the U’s Triangle indicate stress-induced growth modulationPLOS ONE

Dear Dr. Rana,

Thank you for submitting your manuscript to PLOS ONE. After careful consideration, we feel that it has merit but does not fully meet PLOS ONE’s publication criteria as it currently stands. Therefore, we invite you to submit a revised version of the manuscript that addresses the points raised during the review process.

We look forward to receiving your revised manuscript.

Kind regards,

Mojtaba Kordrostami, Ph.D.

Academic Editor

PLOS ONE

Journal Requirements:

Additional Editor Comments:

Dear colleagues,

Thank you for submitting your manuscript entitled "Phylogenomic curation of Ovate Family Proteins (OFPs) in the members of the U’s Triangle indicate stress-induced growth modulation" to PLOS ONE. We have now received and considered the evaluations of your manuscript from three expert reviewers.

After careful consideration, I am pleased to inform you that your manuscript has been deemed suitable for publication following minor revisions. The reviewers have recognized the significance and novelty of your study but have suggested several specific improvements to enhance the clarity and robustness of your manuscript.

Please find below a summary of the key points raised by the reviewers:

Reviewer 1:

Please italicize gene names throughout the text for consistency.

Utilize the Kobas database for the gene ontology section for a more detailed analysis.

Improve the presentation of data in Figure 8 using Graphpad Parsim software and ensure the inclusion of a clear legend for each chart.

Reviewer 2:

Provide evidence (such as gel or nanodrop images) to confirm the quality of the RNA samples used.

Clarify the rationale behind using 1 µg of RNA for transcription in each sample, considering the varying concentrations.

Include details about the fragment size of the targeted genes in the qRT-PCR experiment and the specific PCR conditions used.

Add “Sequence Logos” to demonstrate the conservativeness of the OFP gene family.

Increase the DPI of all images to at least 300 for better clarity.

Recheck and correct the error bars for BnaOFP13 and BnaOFP29 in Figure 8.

Reviewer 3 provided positive feedback and did not suggest any specific revisions.

We kindly ask you to address these comments thoroughly in your revised manuscript. Please ensure that all changes made are highlighted and include a detailed response to each point raised by the reviewers in your resubmission.

We believe that these revisions will significantly strengthen your manuscript and look forward to receiving your revised submission. Should you have any questions or require clarification on any of the points, please do not hesitate to contact us.

Thank you for considering PLOS ONE as a platform for your research. We appreciate the opportunity to consider your work.

Sincerely,

Mojtaba Kordrostami

Editor, PLOS ONE

Reviewers' comments:

Reviewer's Responses to Questions

**Comments to the Author**

1. Is the manuscript technically sound, and do the data support the conclusions?

Reviewer #1: Yes

Reviewer #2: Yes

Reviewer #3: Yes

2. Has the statistical analysis been performed appropriately and rigorously? 

Reviewer #1: Yes

Reviewer #2: Yes

Reviewer #3: Yes

3. Have the authors made all data underlying the findings in their manuscript fully available?

Reviewer #1: No

Reviewer #2: Yes

Reviewer #3: Yes

4. Is the manuscript presented in an intelligible fashion and written in standard English?

Reviewer #1: Yes

Reviewer #2: Yes

Reviewer #3: Yes

5. Review Comments to the Author

Reviewer #1: "Phylogenomic curation of Ovate Family Proteins (OFPs) in the members of the U’s Triangle indicate stress-induced growth modulation" is a very useful study provides a clear overview of the analysis significance and objectives in addressing abiotic stresses in Brassica. It effectively outlines the research focus and the need for developing abiotic cultivars.

1) Please put genes in italics name throughout the text.

2) Please use the Kobas database at http://bioinfo.org/kobas/annotate/ in the gene ontology section because it examines more details as compared to gProfiler.

3) Please use Graphpad Parsim software to show the low value in the graph drawing section in Figure 8. Display the legend for each chart.

Reviewer #2: Material and Method Section

In current study, there are many RNA samples and every RNA sample must have different concentration. Please elaborate how you made sure that the extracted RNA is of good quality. May be gel image of some of the extracted RNA, nanodrop image or concentration will showcase the good quality RNA. Secondly, if RNA every sample have different concentration, then why the authors took 1 µg of RNA to go for transcription? Did you quantify each sample ?

What the fragment size of the targeted genes used in qRT-PCR experiment? And what PCR conditions were used to run the experiment?

The authors claimed OFPs contains conserved domain. It is suggested that “Sequence Logos” may added to further check about the conserveness of the gene family.

Results:

Currently the images are not very clear. It is suggested that DPI of all images need to be upgraded. Each figure/image should have at least 300 DPI.

Error bars in BnaOFP13 and BnaOFP29 (Figure 8) need to be rechecked.

Reviewer #3: The manuscripts entitled “Phylogenomic curation of Ovate Family Proteins (OFPs) in the members of the U’s Triangle indicate stress-induced growth modulation” provides a comprehensive exploration of the Ovate Family Proteins (OFPs) in Brassica species, shedding light on their roles in plant growth and development. The investigation presents a novel and detailed analysis encompassing genetic, regulatory, and functional aspects of this gene family. The authors retrieve and analyze 256 OFP protein sequences, offering a thorough examination of their characteristics. They studied various charecteristics, including chromosomal localization, gene structure, conserved protein motif domains, and the pattern of cis-acting regulatory elements. Notably, the observation of an abundance of light-responsive elements such as G-box, MRE, and GT1 motifs presents intriguing insights into the sensitivity of OFPs to light stimuli. The study enriches its findings by investigating protein-protein interaction networks. Notably, the discovery of the involvement of specific OFPs, especially OFP05 and its orthologous genes, in the process of transcriptional repression through interaction with homeodomain transcription factors like KNAT and BLH is a significant revelation. This exploration is substantiated by the identification of domains like DNA binding 2, hinting at the potential role of OFPs in gene expression regulation.

Moreover, the investigation extends into the realm of biotic and abiotic stresses, showcasing the differential expression patterns of specific OFP genes in response to stress conditions and across various tissues. The highlighted upregulation of genes like BjuOFP30 and BnaOFP27 in specific tissues under stress conditions strongly suggests their pivotal role in plant growth and development processes. The manuscript solidifies its findings by demonstrating experimental validation of BnaOFPs, particularly BnaOFP27, confirming their involvement in regulating gene expression under multiple stress conditions. Furthermore, the manuscript employs GO and KEGG analyses, unveiling the broader roles of OFPs in regulating plant growth and development.

Overall, this manuscript presents a comprehensive and well-executed analysis of the OFP gene family in Brassica, shedding light on their diverse roles in plant growth and stress responses.

6. PLOS authors have the option to publish the peer review history of their article (what does this mean?). If published, this will include your full peer review and any attached files.

Reviewer #1: No

Reviewer #2: **Yes: **Shoaib Ur Rehman

Reviewer #3: **Yes: **Mohammad Aslam

---

## [Author Response · Author response to Decision Letter 0]

3 Jan 2024

To Reviewer #1

Thanks for the positive feedback. We are truly grateful for the insightful revisions that you suggested. We have given each of your comment careful thought and have made the changes to the manuscript accordingly. Our humble response to your comments is given below.

1) Please put genes in italics name throughout the text.

> All the genes names have been edited in italic throughout the manuscript as requested.

2) Please use the Kobas database at http://bioinfo.org/kobas/annotate/ in the gene ontology section because it examines more details as compared to gProfiler.

> Thanks for this suggestion. We realized our shortcomings in the GO & KEGG analysis, so it has been revamped using the more robust KOBAS database as requested. Moreover, the output from the KOBAS-i module have been visualized using the SRplot utility to show the enrichment count of all the significant metabolic pathways. We think that you’ll like the new KOBAS-mediated illustrations. Thanks again…

3) Please use Graphpad Parsim software to show the low value in the graph drawing section in Figure 8. Display the legend for each chart.

> Thanks for the suggestion. All the graphs that were previously developed using Excel have been redeveloped using the GraphPad Prism’s latest version. All the low values have also been highlighted throughout using the numerical view function. All the missing legends have also been added. We hope you’ll like the final results.

To Reviewer #2

Material and Method Section

In current study, there are many RNA samples and every RNA sample must have different concentration. Please elaborate how you made sure that the extracted RNA is of good quality. May be gel image of some of the extracted RNA, nanodrop image or concentration will showcase the good quality RNA. Secondly, if RNA every sample have different concentration, then why the authors took 1 µg of RNA to go for transcription? Did you quantify each sample ?

> Thanks for the comment. You’re absolutely right. Each of the sample have different RNA concentrations and to make sure it’s good quality RNA, we quantified it through a NanoDrop spectrophotometer. These values have now been mentioned in ‘S8 Table’ as desired. Moreover, three technical replicates further improved the quantification outcomes. On the other hand, each sample was quantified separately as mentioned above. In each template for the cDNA synthesis, a 1000ng concentration was kept constant through precise RNA concentration calculations (1000 / x- ngμL-1 of RNA). After that, 2 μL of cDNA was used in the main reaction mixture (2X SYBR Green Master qPCR Mix: 10 μL, F-Primer: 0.5 μL, R-Primer: 0.5 μL, ddH2O: 07 μL, cDNA: 2 μL, Total : 20 μL). We forgot to mention it earlier and now the ‘Material and Methods’ section dealing with qRT-PCR has been modified accordingly. We’re thankful for this valuable insight and hope that you’ll like the modifications.

What the fragment size of the targeted genes used in qRT-PCR experiment? And what PCR conditions were used to run the experiment?

> Thanks for this insight. The fragment sizes of each of the genes have been added in ‘S8 Table’ in front of each of the developed primers. In terms of the subsequent qRT-PCR conditions, the standard manufacturer protocols are as following:

Reverse Transcription (RT):

60°C for 45 minutes.

PCR Amplification:

Initial denaturation: 95°C for 30 seconds.

Denaturation: 95°C for 15 seconds.

Annealing: 60°C for 30 seconds.

Extension: 72°C for 45 seconds.

Final denaturation: 95°C for 30 seconds.

Repeat for 40 cycles.

These conditions and all the other materials that have been used in PCR have been duly mentioned in the manuscript as desired.

The authors claimed OFPs contains conserved domain. It is suggested that “Sequence Logos” may added to further check about the conserveness of the gene family.

> Thanks for the suggestion. We realized that the main image only highlights the conserved domain position so we’ve added a high-quality set of ‘Sequence Logos in the “S1 Dataset” as desired. This will give readers more context about the conserveness of the OVATE domain. We hope you’ll like this addition. Thanks again…

Results:

Currently the images are not very clear. It is suggested that DPI of all images need to be upgraded. Each figure/image should have at least 300 DPI.

> Initially, all the vector images were rendered in 2400 dpi with sizes reaching > 100 Mb. In accordance with the journal submission guidelines, the maximum size is limited to < 10 Mb. Given your suggestion, we’ve reassessed and readjusted all the images for better visualization. It can also be due to the extreme PDF compression in the file that you’re viewing from your point-of-view. We are confident these readjusted images will look better in the final production. Thanks again for the suggestion.

Error bars in BnaOFP13 and BnaOFP29 (Figure 8) need to be rechecked.

> Thanks for the suggestion. We realized this was a typo mistake on our side. We’ve corrected it and rechecked all other data for any other mistakes as well. We have also recreated the graphs in the Figure 8 using the latest version of GraphPad Prism. Even the low values have been numerically highlighted this time for a better experience to the reader. We hope you’ll like the final results. Thanks again.

To Reviewer #3

Dear Reviewer,

We’re truly grateful for all your commentary and thankful that you liked our work. We’ve always strived toward robust researches that bring meaningful outcomes to the field. It feels euphoric when the significance of your hard work is appreciated by a person of the relevent field. Thanks again.

Regards,

Corresponding author

---

## [Decision Letter · Decision Letter 1]

8 Jan 2024

Phylogenomic curation of Ovate Family Proteins (OFPs) in the U’s Triangle of Brassica L. indicates stress-induced growth modulation

PONE-D-23-33855R1

Dear Dr. Rana,

We’re pleased to inform you that your manuscript has been judged scientifically suitable for publication and will be formally accepted for publication once it meets all outstanding technical requirements.

Kind regards,

Mojtaba Kordrostami, Ph.D.

Academic Editor

PLOS ONE

Additional Editor Comments (optional):

The manuscript can be accepted now.

Reviewers' comments:

Reviewer's Responses to Questions

**Comments to the Author**

1. If the authors have adequately addressed your comments raised in a previous round of review and you feel that this manuscript is now acceptable for publication, you may indicate that here to bypass the “Comments to the Author” section, enter your conflict of interest statement in the “Confidential to Editor” section, and submit your "Accept" recommendation.

Reviewer #1: All comments have been addressed

Reviewer #3: All comments have been addressed

2. Is the manuscript technically sound, and do the data support the conclusions?

Reviewer #1: Yes

Reviewer #3: Yes

3. Has the statistical analysis been performed appropriately and rigorously? 

Reviewer #1: Yes

Reviewer #3: Yes

4. Have the authors made all data underlying the findings in their manuscript fully available?

Reviewer #1: No

Reviewer #3: Yes

5. Is the manuscript presented in an intelligible fashion and written in standard English?

Reviewer #1: Yes

Reviewer #3: Yes

6. Review Comments to the Author

Reviewer #1: Phylogenomic curation of Ovate Family Proteins (OFPs) in the members of the U’s Triangle indicate stress-induced growth modulation" is a study provides a clear description of the analysis and objectives in addressing abiotic stresses in Brassica. Thanks to the authors for efforts to respond to comments.

Reviewer #3: (No Response)

7. PLOS authors have the option to publish the peer review history of their article (what does this mean?). If published, this will include your full peer review and any attached files.

Reviewer #1: **Yes: **Zohreh Hajibarat

Reviewer #3: **Yes: **Mohammad Aslam

---

## [Editor Report · Acceptance letter]

17 Jan 2024

PONE-D-23-33855R1 

PLOS ONE

Dear Dr. Rana, 

I'm pleased to inform you that your manuscript has been deemed suitable for publication in PLOS ONE. Congratulations! Your manuscript is now being handed over to our production team.

Kind regards, 

on behalf of

Dr. Mojtaba Kordrostami 

Academic Editor

PLOS ONE